# Some FFT Algorithms for Small-Length Real-Valued Sequences

**Dorota Majorkowska-Mech** \*,† 🄳 and **Aleksandr Cariow** † 🄳

Faculty of Comuter Science and Information Technology, West Pomeranian University of Technology, Zolnierska 49, 71-210 Szczecin, Poland; acariow@wi.zut.edu.pl
* Correspondence: dmajorkowska@wi.zut.edu.pl
† These authors contributed equally to this work.

**Abstract:** This paper proposes fast algorithms for computing the discrete Fourier transform for real-valued sequences of lengths from 3 to 9. Since calculating the real-valued DFT using the complex-valued FFT is redundant regarding the number of needed operations, the developed algorithms do not operate on complex numbers. The algorithms are described in matrix–vector notation and their data flow diagrams are shown.

**Keywords:** discrete Fourier transform; fast algorithm; matrix–vector notation

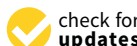



## 1. Introduction

Today, discrete Fourier transform (DFT) is one of the most popular digital signal- and image-processing tools [1–7]. However, it was used quite rarely for a long time due to its high computational complexity. In 1965, J. Cooley and J. Tukey proposed a fast algorithm to compute DFT with a drastically reduced number of arithmetical operations [8]. This discovery caused a furore among specialists and gave impetus to developing high-speed data processing algorithms based on fast Fourier transform (FFT). Mathematically, FFT algorithms are based on a factorization of the Fourier matrix into a product of sparse matrices, meaning matrices with many zero entries. In the case of the Cooley–Tukey algorithm, we are dealing with the representation of the original matrix as a product of $\log_2 N$ sparsely structured matrices. As is well known, the complexity of this algorithm is approximately $(N/2) \log_2 N$ multiplications and the same number of additions of complex numbers.

In DFT algorithms, the input data is usually complex-valued. However, in practical digital signal-processing applications, the dataset for which the DFT is calculated is most often real-valued [9–11]. Therefore, developing algorithms suited explicitly for computing the DFT of real-valued data is an actual problem [12].

DFT algorithms for complex-valued data can be used directly for real-valued data. Specifically, the DFT of a real-valued dataset can be computed by converting it into a complex-valued dataset with zero imaginary parts. This approach, although simple, requires about twice the amount of computation and memory than that required in the case of more efficient algorithms designed for real-valued data. Moreover, ordinarily calculating the real-valued DFT using the complex-valued DFT is redundant regarding the number of operations performed. It is well known that the DFT of a real-valued signal is conjugate-symmetric. Therefore, even intuitively, one can assume that the calculations can be reduced by about half in this case. There are two main rational approaches to using the conventional complex-valued DFT to calculate the real-valued DFT. The first approach allows the simultaneous calculation of two $N$-point real-valued DFTs using a complex-valued DFT of the same size. The second approach is based on the transformation of an $N$-point real-valued sequence into an $N/2$-point complex-valued sequence, which leads to a reduction in computational complexity. Despite the existence of the mentioned approaches, there are a large number of algorithmic solutions developed specifically for real-valued data [12–41]. In this regard, articles [14,25,26] may be particularly noted.

The developed algorithms were presented mainly in algebraic relations, and less often in DFT matrix factorizations. However, none of the publications known to us explains how these ratios were obtained and from what considerations the presented matrices were constructed. This is especially true for small-length real-valued DFT algorithms. Thus, the details of constructing small-length real-valued DFT algorithms are essential. Nevertheless, the solutions given in the literature do not provide a complete imagination about the organization of the small-length real-valued DFT calculation process since their corresponding signal flow graphs are not presented anywhere. In addition, it should be noted that real-valued DFT algorithms for small-length sequences are described in publications available to the authors rather poorly [14].

Applications of short-size FFT algorithms are known from the literature [1,3–5,7]. The need to calculate real-valued DFTs arises, for example, in the world of the Internet of Things, smart sensors and connected devices generate a massive amount of data on petabytes per second. Due to communication costs that impact performance and energy consumption, there is an increasing need to perform a significant amount of computation closer to the edge rather than transferring large portions of raw data to the cloud. The latency and security risk of relying on the cloud is intolerable for applications deployed in drones, autonomous vehicles, robotics, and wearables. These applications are often enabled by DSP algorithms and, more specifically, by small-length real-valued DFTs, which extract meaningful information from raw data. Therefore, the efficient deployment of signal processing in embedded devices will improve near-sensor processing, avoid expensive data transmission, enable freedom from the cloud, and provide low latency and low energy consumption. This is a significant challenge due to the embedded systems resource constraints and the increased computational requirements of data processing.

The purpose of the article is to show the details of the organization of calculations of small-length DFTs for the case of real-valued input data as well as to reduce the number of arithmetic operations needed to compute the output.

Similar solutions for some $N(3,5,7)$ can be found in [14]. However, there are no considerations of how the presented matrices were constructed and the data flow diagrams are not shown. We have completed the missing algorithms for the remaining $N(4,6,8,9)$, described the way in which these algorithms were obtained, and shown the data flow diagrams for all $N$ from 3 to 9.

## 2. Mathematical Background

The DFT of a discrete signal $x(n)$ of size $N$ is given by

$$c(n) = \sum_{k=0}^{N-1} x(k) \exp \frac{-j2\pi nk}{N} = \sum_{k=0}^{N-1} x(k) \left( \cos \frac{2\pi nk}{N} - j \sin \frac{2\pi nk}{N} \right) \tag{1}$$

for $n = 0, 1, \ldots, N-1$, where $j$ is the imaginary unit. Each coefficient $c(n)$ of DFT, as a complex number, can be written as a sum

$$c(n) = a(n) + jb(n) \tag{2}$$

where $a(n)$ and $b(n)$ are the a real part and an imaginary part of $c(n)$, respectively. For real signal $x(n)$

$$a(n) = \sum_{k=0}^{N-1} x(k) \cos \frac{2\pi nk}{N} \tag{3}$$

$$b(n) = -\sum_{k=0}^{N-1} x(k) \sin \frac{2\pi nk}{N} \tag{4}$$

In this case, the DFT is redundant since it is conjugate-symmetric. Its real and imaginary parts meet the following properties:

$$a(N-n) = \sum_{k=0}^{N-1} x(k) \cos \frac{2\pi(N-n)k}{N} = \sum_{k=0}^{N-1} x(k) \cos \frac{2\pi nk}{N} = a(n) \tag{5}$$

$$b(N-n) = -\sum_{k=0}^{N-1} x(k) \sin \frac{2\pi(N-n)k}{N} = \sum_{k=0}^{N-1} x(k) \sin \frac{2\pi nk}{N} = -b(n) \tag{6}$$

Taking into account the periodicity of the signal $x(n)$ and their DFT with the period $N$, we can write

$$a(N - n) = a(-n) = a(n) \tag{7}$$

so the real part of DFT is an even function. Since

$$b(N - n) = b(-n) = -b(n) \tag{8}$$

then the imaginary part of DFT is an odd function. From (8) for $n = 0$ we have $b(-0) = b(0) = -b(0)$, so we obtain

$$b(0) = 0 \tag{9}$$

and if $N$ is an even number $b(N/2) = b(N-N/2) = b(-N/2) = -b(N/2)$, so

$$b\left(\frac{N}{2}\right) = 0 \tag{10}$$

In order to avoid redundancy and to use real transform for real signals the real discrete Fourier transform (RDFT) has been defined [25]

$$y(n) = \sum_{k=0}^{N-1} x(k) \cos\left(\frac{2\pi nk}{N} + \theta(n)\right) \tag{11}$$

where

$$\theta(n) = \begin{cases} 0 & 0 \leq n \leq \lfloor \frac{N}{2} \rfloor \\ \frac{\pi}{2} & \lfloor \frac{N}{2} \rfloor < n \leq N - 1 \end{cases} \tag{12}$$

and $\lfloor \cdot \rfloor$ denotes the floor function.

To see the relationship between the RDFT and the DFT of the signal $x(n)$, it is better to write the RDFT in a slightly different form

$$y(n) = \begin{cases} \sum_{k=0}^{N-1} x(k) \cos \frac{2\pi nk}{N} & 0 \leq n \leq \lfloor \frac{N}{2} \rfloor \\ -\sum_{k=0}^{N-1} x(k) \sin \frac{2\pi nk}{N} & \lfloor \frac{N}{2} \rfloor < n \leq N - 1 \end{cases} \tag{13}$$

Now it is clear that

$$y(n) = \begin{cases} a(n) & 0 \leq n \leq \lfloor \frac{N}{2} \rfloor \\ b(n) & \lfloor \frac{N}{2} \rfloor < n \leq N - 1 \end{cases} \tag{14}$$

so the RDFT contains the not repeated and non-zero coefficients from the real and imaginary parts of the DFT.

Knowing the RDFT coefficients of the signal and the relationships (7)–(10) and (14), it is easy to obtain the DFT coefficients. Namely, if $N$ is an even number then

$$
\begin{aligned}
&a(0) = y(0) && b(0) = 0 \\
&a(1) = y(1) && b(1) = -b(N-1) = -y(N-1) \\
&\quad\vdots && \quad\vdots \\
&a\left(\tfrac{N}{2}-1\right) = y\left(\tfrac{N}{2}-1\right) && b\left(\tfrac{N}{2}-1\right) = -b\left(\tfrac{N}{2}+1\right) = -y\left(\tfrac{N}{2}+1\right) \\
&a\left(\tfrac{N}{2}\right) = y\left(\tfrac{N}{2}\right) && b\left(\tfrac{N}{2}\right) = 0 \\
&a\left(\tfrac{N}{2}+1\right) = a\left(\tfrac{N}{2}-1\right) = y\left(\tfrac{N}{2}-1\right) && b\left(\tfrac{N}{2}+1\right) = y\left(\tfrac{N}{2}+1\right) \\
&\quad\vdots && \quad\vdots \\
&a(N-1) = a(1) = y(1) && b(N-1) = y(N-1)
\end{aligned}
\tag{15}
$$

If $N$ is an odd number then

$$
\begin{aligned}
&a(0) = y(0) && b(0) = 0 \\
&a(1) = y(1) && b(1) = -b(N-1) = -y(N-1) \\
&\quad\vdots && \quad\vdots \\
&a\left(\tfrac{N-1}{2}\right) = y\left(\tfrac{N-1}{2}\right) && b\left(\tfrac{N-1}{2}\right) = -b\left(\tfrac{N+1}{2}\right) = -y\left(\tfrac{N+1}{2}\right) \\
&a\left(\tfrac{N+1}{2}\right) = a\left(\tfrac{N-1}{2}\right) = y\left(\tfrac{N-1}{2}\right) && b\left(\tfrac{N+1}{2}\right) = y\left(\tfrac{N+1}{2}\right) \\
&\quad\vdots && \quad\vdots \\
&a(N-1) = a(1) = y(1) && b(N-1) = y(N-1)
\end{aligned}
\tag{16}
$$

Later in the article, the matrix–vector notation will be used, so the discrete real input signal $x(n)$ of size $N$ will be represented by a column vector $\mathbf{x}_N = [x_0, x_1, \ldots, x_{N-1}]^T$, the $N$-by-$N$ RDFT matrix will be denoted by $\mathbf{R}_N$ and the output signal $y(n)$—by $\mathbf{y}_N = [y_0, y_1, \ldots, y_{N-1}]^T$, where $[\cdot]^T$ means standard transposition operation. In matrix–vector notation the RDFT transform can be described as follows:

$$
\mathbf{y}_N = \mathbf{R}_N \mathbf{x}_N
\tag{17}
$$

The entries $r_{nk}$ of the matrix $\mathbf{R}_N$ are obtained from (11)

$$
r_{nk} = \cos\left(\frac{2\pi nk}{N} + \theta_n\right)
\tag{18}
$$

where the indexes $n$ and $k$ vary from 0 to $N-1$ and $\theta_n = \theta(n)$ is defined by (12).

### 3. RDFT Algorithm for $N = 3$

For $N = 2$, the DFT is real-valued for real input signal and it is the same as RDFT, so we will start from $N = 3$. We introduce the denotation $\phi_N = 2\pi/N$. In this case the Equation (17) will take the form

$$
\mathbf{y}_3 = \mathbf{R}_3 \mathbf{x}_3
\tag{19}
$$

where

$$
\mathbf{R}_3 =
\begin{bmatrix}
\cos 0 & \cos 0 & \cos 0 \\
\cos 0 & \cos \phi_3 & \cos 2\phi_3 \\
\cos \frac{\pi}{2} & \cos\left(2\phi_3 + \frac{\pi}{2}\right) & \cos\left(4\phi_3 + \frac{\pi}{2}\right)
\end{bmatrix}
=
\begin{bmatrix}
1 & 1 & 1 \\
1 & \cos \phi_3 & \cos 2\phi_3 \\
0 & -\sin 2\phi_3 & -\sin 4\phi_3
\end{bmatrix}
\tag{20}
$$

Since $\phi_3 = 2\pi/3$ then using the trigonometric reduction formulas we obtain $\cos 2\phi_3 = \cos \phi_3$, $\sin 2\phi_3 = -\sin \phi_3$, and $\sin 4\phi_3 = \sin \phi_3$, so the $\mathbf{R}_3$ matrix will take the form

$$
\mathbf{R}_3 =
\begin{bmatrix}
1 & 1 & 1 \\
1 & \cos \phi_3 & \cos \phi_3 \\
0 & \sin \phi_3 & -\sin \phi_3
\end{bmatrix}
\tag{21}
$$

When we calculate the product of this matrix by the input vector $\mathbf{x}_3$ we obtain

$$\begin{bmatrix} y_0 \\ y_1 \\ y_2 \end{bmatrix} = \begin{bmatrix} x_0 + (x_1 + x_2) \\ x_0 + \cos\phi_3(x_1 + x_2) \\ \sin\phi_3(x_1 - x_2) \end{bmatrix} = \begin{bmatrix} x_0 + (x_1 + x_2) \\ (\cos\phi_3 - 1)(x_1 + x_2) + [x_0 + (x1 + x_2)] \\ \sin\phi_3(x_1 - x_2) \end{bmatrix} \tag{22}$$

Figure 1 shows a data flow diagram corresponding to this calculation, where $d_1 = \cos\phi_3 - 1$ and $d_2 = \sin\phi_3$.

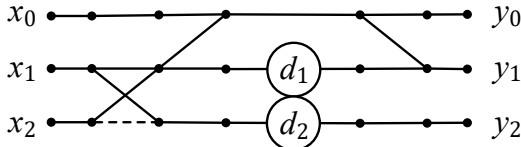

**Figure 1.** Data-flow diagram of the RDFT algorithm for $N = 3$.

In this paper, data flow diagrams are oriented from left to right. Straight lines in the figures denote the operations of data transfer. Points where lines converge denote summation. The dotted lines indicate a signed-changed data transfer operation. The circles in these figures show the operation of multiplication by a number inscribed inside a circle.

When the vector $\mathbf{z}_3 = \mathbf{a}_3 + i\mathbf{b}_3$ of complex coefficients of DFT for the real input vector $\mathbf{x}_3$ is needed, it can be easily obtained from the output vector $\mathbf{y}_3$ of RDFT, according to (16)

$$\mathbf{a}_3 = \begin{bmatrix} a_0 \\ a_1 \\ a_2 \end{bmatrix} = \begin{bmatrix} y_0 \\ y_1 \\ y_1 \end{bmatrix} \quad \mathbf{b}_3 = \begin{bmatrix} b_0 \\ b_1 \\ b_2 \end{bmatrix} = \begin{bmatrix} 0 \\ -y_2 \\ y_2 \end{bmatrix} \tag{23}$$

The algorithm of the RDFT for $N = 3$, presented in Figure 1, can be described by the following matrix–vector procedure, in which the matrix $\mathbf{R}_3$ has been factorized:

$$\mathbf{y}_3 = \mathbf{C}_3\mathbf{D}_3\tilde{\mathbf{A}}_3\hat{\mathbf{A}}_3\mathbf{y}_3 \tag{24}$$

where

$$\hat{\mathbf{A}}_3 = \begin{bmatrix} 1 & 0 & 0 \\ 0 & 1 & 1 \\ 0 & 1 & -1 \end{bmatrix} \quad \tilde{\mathbf{A}}_3 = \begin{bmatrix} 1 & 1 & 0 \\ 0 & 1 & 0 \\ 0 & 0 & 1 \end{bmatrix} \quad \mathbf{D}_3 = \begin{bmatrix} 1 & 0 & 0 \\ 0 & d_1 & 0 \\ 0 & 0 & d_2 \end{bmatrix} \quad \mathbf{C}_3 = \begin{bmatrix} 1 & 0 & 0 \\ 1 & 1 & 0 \\ 0 & 0 & 1 \end{bmatrix} \tag{25}$$

According to this algorithm we need only 2 multiplications and 4 additions of real numbers to calculate the output vector $\mathbf{y}_3$. It should be noted that only the matrix $\mathbf{D}_3$ is responsible for multiplications and that matrix is diagonal

$$\mathbf{D}_3 = \text{diag}(1, d_1, d_2) \tag{26}$$

## 4. RDFT Algorithm for $N = 4$

For $N = 4$, the Equation (17) will take the form

$$\mathbf{y}_4 = \mathbf{R}_4\mathbf{x}_4 \tag{27}$$

where

$$\mathbf{R}_4 = \begin{bmatrix} 1 & 1 & 1 & 1 \\ 1 & \cos\phi_4 & \cos 2\phi_4 & \cos 3\phi_4 \\ 1 & \cos 2\phi_4 & \cos 4\phi_4 & \cos 6\phi_4 \\ 0 & -\sin 3\phi_4 & -\sin 6\phi_4 & -\sin 9\phi_4 \end{bmatrix} \tag{28}$$

Since $\phi_4 = 2\pi/4 = \pi/2$ then $\cos\phi_4 = 0$, $\cos 2\phi_4 = -1$, $\cos 3\phi_4 = 0$, $\cos 4\phi_4 = 1$, $\cos 6\phi_4 = -1$ and $\sin 3\phi_4 = -1$, $\sin 6\phi_4 = 0$, $\sin 9\phi_4 = 1$, so the $\mathbf{R}_4$ matrix will take the form

$$\mathbf{R}_4 = \begin{bmatrix} 1 & 1 & 1 & 1 \\ 1 & 0 & -1 & 0 \\ 1 & -1 & 1 & -1 \\ 0 & 1 & 0 & -1 \end{bmatrix} \tag{29}$$

When we calculate the product of this matrix by the real input vector $\mathbf{x}_4$ we obtain

$$\begin{bmatrix} y_0 \\ y_1 \\ y_2 \\ y_3 \end{bmatrix} = \begin{bmatrix} (x_0 + x_2) + (x_1 + x_3) \\ x_0 - x_2 \\ (x_0 + x_2) - (x_1 + x_3) \\ x_1 - x_3 \end{bmatrix} \tag{30}$$

Figure 2 shows a data-flow diagram corresponding to this calculation.

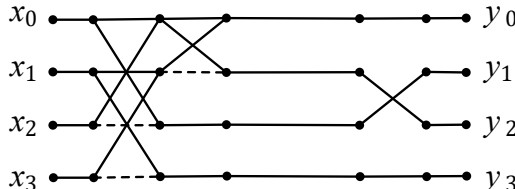

**Figure 2.** Data-flow diagram of the RDFT algorithm for $N = 4$.

When the vector $\mathbf{z}_4 = \mathbf{a}_4 + i\mathbf{b}_4$ of complex coefficients of DFT for the input vector $\mathbf{x}_4$ is needed, it can be easily obtained from the output vector $\mathbf{y}_4$ of RDFT, according to (15)

$$\mathbf{a}_4 = \begin{bmatrix} a_0 \\ a_1 \\ a_2 \\ a_3 \end{bmatrix} = \begin{bmatrix} y_0 \\ y_1 \\ y_2 \\ y_1 \end{bmatrix} \qquad \mathbf{b}_4 = \begin{bmatrix} b_0 \\ b_1 \\ b_2 \\ b_3 \end{bmatrix} = \begin{bmatrix} 0 \\ -y_3 \\ 0 \\ y_3 \end{bmatrix} \tag{31}$$

The algorithm of the RDFT for $N = 4$, presented in Figure 2, can be described by the following matrix-vector procedure, in which the matrix $\mathbf{R}_4$ has been factorized:

$$\mathbf{y}_4 = \mathbf{C}_4\mathbf{D}_4\tilde{\mathbf{A}}_4\hat{\mathbf{A}}_4\mathbf{x}_4 \tag{32}$$

where

$$\hat{\mathbf{A}}_4 = \begin{bmatrix} 1 & 0 & 1 & 0 \\ 0 & 1 & 0 & 1 \\ 1 & 0 & -1 & 0 \\ 0 & 1 & 0 & -1 \end{bmatrix} \quad \tilde{\mathbf{A}}_4 = \begin{bmatrix} 1 & 1 & 0 & 0 \\ 1 & -1 & 0 & 0 \\ 0 & 0 & 1 & 0 \\ 0 & 0 & 0 & 1 \end{bmatrix} \quad \mathbf{D}_4 = \mathbf{I}_4 \quad \mathbf{C}_4 = \begin{bmatrix} 1 & 0 & 0 & 0 \\ 0 & 0 & 1 & 0 \\ 0 & 1 & 0 & 0 \\ 0 & 0 & 0 & 1 \end{bmatrix} \tag{33}$$

and $\mathbf{I}_k$ denotes the identity matrix of size $k$.

According to this algorithm, we need only 6 additions of real numbers to calculate the output vector $\mathbf{y}_4$. It should be noted that the matrix $\mathbf{D}_4$ is responsible for multiplications and that, in this case, this matrix is equal to identity matrix, so we need not any multiplications of real numbers.

## 5. RDFT Algorithm for $N = 5$

For $N = 5$, the Equation (17) will take the form

$$\mathbf{y}_5 = \mathbf{R}_5\mathbf{x}_5 \tag{34}$$

where

$$
\mathbf{R}_5 =
\begin{bmatrix}
1 & 1 & 1 & 1 & 1 \\
1 & \cos\phi_5 & \cos 2\phi_5 & \cos 3\phi_5 & \cos 4\phi_5 \\
1 & \cos 2\phi_5 & \cos 4\phi_5 & \cos 6\phi_5 & \cos 8\phi_5 \\
0 & -\sin 3\phi_5 & -\sin 6\phi_5 & -\sin 9\phi_5 & -\sin 12\phi_5 \\
0 & -\sin 4\phi_5 & -\sin 8\phi_5 & -\sin 12\phi_5 & -\sin 16\phi_5
\end{bmatrix}
\tag{35}
$$

Since $\phi_5 = 2\pi/5$ then $\cos 3\phi_5 = \cos 2\phi_5$, $\cos 4\phi_5 = \cos\phi_5$, $\cos 6\phi_5 = \cos\phi_5$, $\cos 8\phi_5 = \cos 2\phi_5$ and $\sin 3\phi_5 = -\sin 2\phi_5$, $\sin 4\phi_5 = -\sin\phi_5$, $\sin 6\phi_5 = \sin\phi_5$, $\sin 8\phi_5 = -\sin 2\phi_5$, $\sin 9\phi_5 = -\sin\phi_5$, $\sin 12\phi_5 = \sin 2\phi_5$, $\sin 16\phi_5 = \sin\phi_5$, so the $\mathbf{R}_5$ matrix will take the form

$$
\mathbf{R}_5 =
\begin{bmatrix}
1 & 1 & 1 & 1 & 1 \\
1 & \cos\phi_5 & \cos 2\phi_5 & \cos 2\phi_5 & \cos\phi_5 \\
1 & \cos 2\phi_5 & \cos\phi_5 & \cos\phi_5 & \cos 2\phi_5 \\
0 & \sin 2\phi_5 & -\sin\phi_5 & \sin\phi_5 & -\sin 2\phi_5 \\
0 & \sin\phi_5 & \sin 2\phi_5 & -\sin 2\phi_5 & -\sin\phi_5
\end{bmatrix}
\tag{36}
$$

When we calculate the product of this matrix by the input vector $\mathbf{x}_5$ we obtain

$$
\begin{bmatrix}
y_0 \\ y_1 \\ y_2 \\ y_3 \\ y_4
\end{bmatrix}
=
\begin{bmatrix}
x_0 + [(x_1 + x_4) + (x_2 + x_3)] \\
x_0 + \cos\phi_5(x_1 + x_4) + \cos 2\phi_5(x_2 + x_3) \\
x_0 + \cos 2\phi_5(x_1 + x_4) + \cos\phi_5(x_2 + x_3) \\
\sin 2\phi_5(x_1 - x_4) - \sin\phi_5(x_2 - x_3) \\
\sin\phi_5(x_1 - x_4) + \sin 2\phi_5(x_2 - x_3)
\end{bmatrix}
\tag{37}
$$

To better understand the construction of the RDFT algorithm for $N = 5$, we will introduce the notations $t_1 = x_1 + x_4$, $t_2 = x_2 + x_3$, $t_3 = x_2 - x_3$, $t_4 = x_1 - x_4$, $t_0 = x_0 + t_1 + t_2$ and consider the sub-blocks of the output vector $\mathbf{y}_5$. The first sub-block is

$$
\begin{bmatrix} y_1 \\ y_2 \end{bmatrix}
=
\begin{bmatrix} x_0 \\ x_0 \end{bmatrix}
+
\begin{bmatrix} \cos\phi_5 & \cos 2\phi_5 \\ \cos 2\phi_5 & \cos\phi_5 \end{bmatrix}
\begin{bmatrix} t_1 \\ t_2 \end{bmatrix}
=
$$

$$
=
\begin{bmatrix} x_0 \\ x_0 \end{bmatrix}
+
\begin{bmatrix} 1 & 1 \\ 1 & -1 \end{bmatrix}
\begin{bmatrix}
\dfrac{\cos\phi_5 + \cos 2\phi_5}{2} & 0 \\
0 & \dfrac{\cos\phi_5 - \cos 2\phi_5}{2}
\end{bmatrix}
\begin{bmatrix} 1 & 1 \\ 1 & -1 \end{bmatrix}
\begin{bmatrix} t_1 \\ t_2 \end{bmatrix}
\tag{38}
$$

We can also write it as

$$
\begin{bmatrix} y_1 \\ y_2 \end{bmatrix}
=
\begin{bmatrix} t_0 \\ t_0 \end{bmatrix}
+
\begin{bmatrix} 1 & 1 \\ 1 & -1 \end{bmatrix}
\begin{bmatrix}
\dfrac{\cos\phi_5 + \cos 2\phi_5}{2} - 1 & 0 \\
0 & \dfrac{\cos\phi_5 - \cos 2\phi_5}{2}
\end{bmatrix}
\begin{bmatrix} 1 & 1 \\ 1 & -1 \end{bmatrix}
\begin{bmatrix} t_1 \\ t_2 \end{bmatrix}
\tag{39}
$$

The second sub-block is

$$
\begin{bmatrix} y_3 \\ y_4 \end{bmatrix}
=
\begin{bmatrix} -\sin\phi_5 & \sin 2\phi_5 \\ \sin 2\phi_5 & \sin\phi_5 \end{bmatrix}
\begin{bmatrix} t_3 \\ t_4 \end{bmatrix}
=
$$

$$
=
\begin{bmatrix} 1 & 0 & 1 \\ 0 & 1 & 1 \end{bmatrix}
\begin{bmatrix}
-\sin\phi_5 - \sin 2\phi_5 & 0 & 0 \\
0 & \sin\phi_5 - \sin 2\phi_5 & 0 \\
0 & 0 & \sin 2\phi_5
\end{bmatrix}
\begin{bmatrix} 1 & 0 \\ 0 & 1 \\ 1 & 1 \end{bmatrix}
\begin{bmatrix} t_3 \\ t_4 \end{bmatrix}
\tag{40}
$$

Figure 3 shows a data flow diagram corresponding to the calculation of the output vector, where $d_1 = (\cos\phi_5 + \cos 2\phi_5)/2 - 1$, $d_2 = (\cos\phi_5 - \cos 2\phi_5)/2$, $d_3 = -\sin\phi_5 - \sin 2\phi_5$, $d_4 = \sin\phi_5 - \sin 2\phi_5$ and $d_5 = \sin 2\phi_5$.

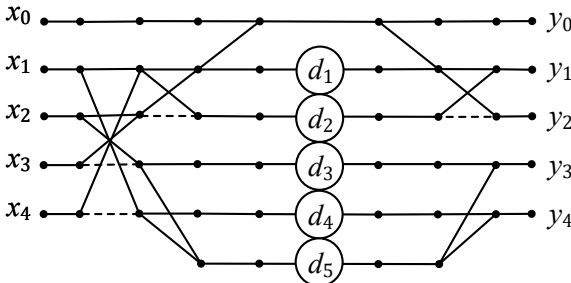

**Figure 3.** Data-flow diagram of the RDFT algorithm for $N = 5$.

When the vector $\mathbf{z}_5 = \mathbf{a}_5 + i\mathbf{b}_5$ of complex coefficients of DFT for the real input vector $\mathbf{x}_5$ is needed, it can be easily obtained from the output vector $\mathbf{y}_5$ of RDFT, according to (16)

$$\mathbf{a}_5 = \begin{bmatrix} a_0 \\ a_1 \\ a_2 \\ a_3 \\ a_4 \end{bmatrix} = \begin{bmatrix} y_0 \\ y_1 \\ y_2 \\ y_2 \\ y_1 \end{bmatrix} \qquad \mathbf{b}_5 = \begin{bmatrix} b_0 \\ b_1 \\ b_2 \\ b_3 \\ b_4 \end{bmatrix} = \begin{bmatrix} 0 \\ -y_4 \\ -y_3 \\ y_3 \\ y_4 \end{bmatrix} \tag{41}$$

The algorithm of the RDFT for $N = 5$, presented in Figure 3, can be described by the following matrix–vector procedure, in which the matrix $\mathbf{R}_5$ has been factorized:

$$\mathbf{y}_5 = \mathbf{C}_{5\times 6}\tilde{\mathbf{C}}_6\mathbf{D}_6\tilde{\mathbf{A}}_6\overline{\mathbf{A}}_{6\times 5}\hat{\mathbf{A}}_5\mathbf{x}_5 \tag{42}$$

where

$$\hat{\mathbf{A}}_5 = \begin{bmatrix} 1 & 0 & 0 & 0 & 0 \\ 0 & 1 & 0 & 0 & 1 \\ 0 & 0 & 1 & 1 & 0 \\ 0 & 0 & 1 & -1 & 0 \\ 0 & 1 & 0 & 0 & -1 \end{bmatrix} \quad \overline{\mathbf{A}}_{6\times 5} = \begin{bmatrix} 1 & 0 & 0 & 0 & 0 \\ 0 & 1 & 1 & 0 & 0 \\ 0 & 1 & -1 & 0 & 0 \\ 0 & 0 & 0 & 1 & 0 \\ 0 & 0 & 0 & 0 & 1 \\ 0 & 0 & 0 & 1 & 1 \end{bmatrix} \quad \tilde{\mathbf{A}}_6 = \begin{bmatrix} 1 & 1 & 0 & 0 & 0 & 0 \\ 0 & 1 & 0 & 0 & 0 & 0 \\ 0 & 0 & 1 & 0 & 0 & 0 \\ 0 & 0 & 0 & 1 & 0 & 0 \\ 0 & 0 & 0 & 0 & 1 & 0 \\ 0 & 0 & 0 & 0 & 0 & 1 \end{bmatrix} \tag{43}$$

$$\mathbf{D}_6 = \begin{bmatrix} 1 & 0 & 0 & 0 & 0 & 0 \\ 0 & d_1 & 0 & 0 & 0 & 0 \\ 0 & 0 & d_2 & 0 & 0 & 0 \\ 0 & 0 & 0 & d_3 & 0 & 0 \\ 0 & 0 & 0 & 0 & d_4 & 0 \\ 0 & 0 & 0 & 0 & 0 & d_5 \end{bmatrix} \quad \tilde{\mathbf{C}}_6 = \begin{bmatrix} 1 & 0 & 0 & 0 & 0 & 0 \\ 1 & 1 & 0 & 0 & 0 & 0 \\ 0 & 0 & 1 & 0 & 0 & 0 \\ 0 & 0 & 0 & 1 & 0 & 0 \\ 0 & 0 & 0 & 0 & 1 & 0 \\ 0 & 0 & 0 & 0 & 0 & 1 \end{bmatrix} \quad \mathbf{C}_{5\times 6} = \begin{bmatrix} 1 & 0 & 0 & 0 & 0 & 0 \\ 0 & 1 & 1 & 0 & 0 & 0 \\ 0 & 1 & -1 & 0 & 0 & 0 \\ 0 & 0 & 0 & 1 & 0 & 1 \\ 0 & 0 & 0 & 0 & 1 & 1 \end{bmatrix} \tag{44}$$

According to this algorithm we need only 13 additions and 5 multiplications of real numbers to calculate the output vector $\mathbf{y}_5$.

## 6. RDFT Algorithm for $N = 6$

For $N = 6$, the Equation (17) will take the form

$$\mathbf{y}_6 = \mathbf{R}_6\mathbf{x}_6 \tag{45}$$

where

$$\mathbf{R}_6 = \begin{bmatrix} 1 & 1 & 1 & 1 & 1 & 1 \\ 1 & \cos\phi_6 & \cos 2\phi_6 & \cos 3\phi_6 & \cos 4\phi_6 & \cos 5\phi_6 \\ 1 & \cos 2\phi_6 & \cos 4\phi_6 & \cos 6\phi_6 & \cos 8\phi_6 & \cos 10\phi_6 \\ 1 & \cos 3\phi_6 & \cos 6\phi_6 & \cos 9\phi_6 & \cos 12\phi_6 & \cos 15\phi_6 \\ 0 & -\sin 4\phi_6 & -\sin 8\phi_6 & -\sin 12\phi_6 & -\sin 16\phi_6 & -\sin 20\phi_6 \\ 0 & -\sin 5\phi_6 & -\sin 10\phi_6 & -\sin 15\phi_6 & -\sin 20\phi_6 & -\sin 25\phi_6 \end{bmatrix} \tag{46}$$

Since $\phi_6 = 2\pi/6$ then $\cos 2\phi_6 = -\cos\phi_6$, $\cos 3\phi_6 = -1$, $\cos 4\phi_6 = -\cos\phi_6$, $\cos 5\phi_6 = \cos\phi_6$, $\cos 6\phi_6 = 1$, $\cos 8\phi_6 = -\cos\phi_6$, $\cos 9\phi_6 = -1$, $\cos 10\phi_6 = -\cos\phi_6$, $\cos 12\phi_6 = 1$, $\cos 15\phi_6 = -1$ and $\sin 4\phi_6 = -\sin\phi_6$, $\sin 5\phi_6 = -\sin\phi_6$, $\sin 8\phi_6 = \sin\phi_6$, $\sin 10\phi_6 = -\sin\phi_6$, $\sin 12\phi_6 = 0$, $\sin 15\phi_6 = 0$, $\sin 16\phi_6 = -\sin\phi_6$, $\sin 20\phi_6 = \sin\phi_6$, $\sin 25\phi_6 = \sin\phi_6$, so the $\mathbf{R}_6$ matrix will take the form

$$\mathbf{R}_6 = \begin{bmatrix} 1 & 1 & 1 & 1 & 1 & 1 \\ 1 & \cos\phi_6 & -\cos\phi_6 & -1 & -\cos\phi_6 & \cos\phi_6 \\ 1 & -\cos\phi_6 & -\cos\phi_6 & 1 & -\cos\phi_6 & -\cos\phi_6 \\ 1 & -1 & 1 & -1 & 1 & -1 \\ 0 & \sin\phi_6 & -\sin\phi_6 & 0 & \sin\phi_6 & -\sin\phi_6 \\ 0 & \sin\phi_6 & \sin\phi_6 & 0 & -\sin\phi_6 & -\sin\phi_6 \end{bmatrix} \tag{47}$$

When we calculate the product of this matrix by the input vector $\mathbf{x}_6$ we obtain

$$\begin{bmatrix} y_0 \\ y_1 \\ y_2 \\ y_3 \\ y_4 \\ y_5 \end{bmatrix} = \begin{bmatrix} (x_0 + x_3) + (x_1 + x_5) + (x_2 + x_4) \\ (x_0 - x_3) + \cos\phi_6[(x_1 + x_5) - (x_2 + x_4)] \\ (x_0 + x_3) + \cos\phi_6[-(x_1 + x_5) - (x_2 + x_4)] \\ (x_0 - x_3) - (x_1 + x_5) + (x_2 + x_4) \\ \sin\phi_6[(x_1 - x_5) - (x_2 - x_4)] \\ \sin\phi_6[(x_1 - x_5) + (x_2 - x_4)] \end{bmatrix} =$$

$$= \begin{bmatrix} (x_0 + x_3) + [(x_1 + x_5) + (x_2 + x_4)] \\ (\cos\phi_6 + 1)[(x_1 + x_5) - (x_2 + x_4)] + [(x_0 - x_3) - (x_1 + x_5) + (x_2 + x_4)] \\ (-\cos\phi_6 - 1)[(x_1 + x_5) + (x_2 + x_4)] + [(x_0 + x_3) + (x_1 + x_5) + (x_2 + x_4)] \\ (x_0 - x_3) - (x_1 + x_5) + (x_2 + x_4) \\ -\sin\phi_6[(x_2 - x_4) - (x_1 - x_5)] \\ \sin\phi_6[(x_2 - x_4) + (x_1 - x_5)] \end{bmatrix} \tag{48}$$

Figure 4 shows a data-flow diagram corresponding to the calculation of the output vector, where $d_1 = -\cos\phi_6 - 1$, $d_2 = \cos\phi_6 + 1$, $d_4 = -\sin\phi_6$, and $d_5 = \sin\phi_6$.

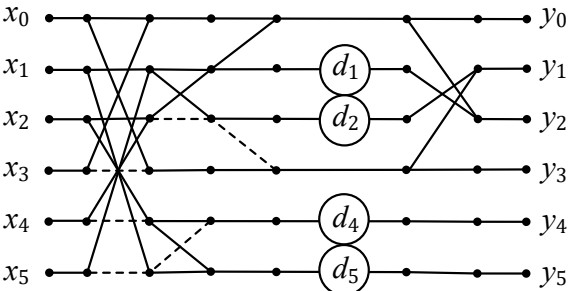

**Figure 4.** Data flow diagram of the RDFT algorithm for $N = 6$.

When the vector $\mathbf{z}_6 = \mathbf{a}_6 + i\mathbf{b}_6$ of complex coefficients of DFT for the real input vector $\mathbf{x}_6$ is needed, it can be easily obtained from the output vector $\mathbf{y}_6$ of RDFT, according to (15)

$$\mathbf{a}_6 = \begin{bmatrix} a_0 \\ a_1 \\ a_2 \\ a_3 \\ a_4 \\ a_5 \end{bmatrix} = \begin{bmatrix} y_0 \\ y_1 \\ y_2 \\ y_3 \\ y_2 \\ y_1 \end{bmatrix} \qquad \mathbf{b}_6 = \begin{bmatrix} b_0 \\ b_1 \\ b_2 \\ b_3 \\ b_4 \\ b_5 \end{bmatrix} = \begin{bmatrix} 0 \\ -y_5 \\ -y_4 \\ 0 \\ y_4 \\ y_5 \end{bmatrix} \tag{49}$$

The algorithm of the RDFT for $N = 6$, presented in Figure 4, can be described by the following matrix-vector procedure, in which the matrix $\mathbf{R}_6$ has been factorized:

$$\mathbf{y}_6 = \mathbf{C}_6 \mathbf{D}_6 \tilde{\mathbf{A}}_6 \overline{\mathbf{A}}_6 \hat{\mathbf{A}}_6 \mathbf{x}_6 \tag{50}$$

where

$$\hat{\mathbf{A}}_6 = \begin{bmatrix} 1 & 0 & 0 & 1 & 0 & 0 \\ 0 & 1 & 0 & 0 & 0 & 1 \\ 0 & 0 & 1 & 0 & 1 & 0 \\ 1 & 0 & 0 & -1 & 0 & 0 \\ 0 & 0 & 1 & 0 & -1 & 0 \\ 0 & 1 & 0 & 0 & 0 & -1 \end{bmatrix} \quad \overline{\mathbf{A}}_6 = \begin{bmatrix} 1 & 0 & 0 & 0 & 0 & 0 \\ 0 & 1 & 1 & 0 & 0 & 0 \\ 0 & 1 & -1 & 0 & 0 & 0 \\ 0 & 0 & 0 & 1 & 0 & 0 \\ 0 & 0 & 0 & 0 & 1 & -1 \\ 0 & 0 & 0 & 0 & 1 & 1 \end{bmatrix} \tag{51}$$

$$\tilde{\mathbf{A}}_6 = \begin{bmatrix} 1 & 1 & 0 & 0 & 0 & 0 \\ 0 & 1 & 0 & 0 & 0 & 0 \\ 0 & 0 & 1 & 0 & 0 & 0 \\ 0 & 0 & -1 & 1 & 0 & 0 \\ 0 & 0 & 0 & 0 & 1 & 0 \\ 0 & 0 & 0 & 0 & 0 & 1 \end{bmatrix} \quad \mathbf{D}_6 = \begin{bmatrix} 1 & 0 & 0 & 0 & 0 & 0 \\ 0 & d_1 & 0 & 0 & 0 & 0 \\ 0 & 0 & d_2 & 0 & 0 & 0 \\ 0 & 0 & 0 & 1 & 0 & 0 \\ 0 & 0 & 0 & 0 & d_4 & 0 \\ 0 & 0 & 0 & 0 & 0 & d_5 \end{bmatrix} \tag{52}$$

$$\mathbf{C}_6 = \begin{bmatrix} 1 & 0 & 0 & 0 & 0 & 0 \\ 0 & 0 & 1 & 1 & 0 & 0 \\ 1 & 1 & 0 & 0 & 0 & 0 \\ 0 & 0 & 0 & 1 & 0 & 0 \\ 0 & 0 & 0 & 0 & 1 & 0 \\ 0 & 0 & 0 & 0 & 0 & 1 \end{bmatrix} \tag{53}$$

According to this algorithm we need only 14 additions and 4 multiplications of real numbers to calculate the output vector $\mathbf{y}_6$.

## 7. RDFT Algorithm for $N = 7$

For $N = 7$, the Equation (17) will take the form

$$\mathbf{y}_7 = \mathbf{R}_7 \mathbf{x}_7 \tag{54}$$

where

$$\mathbf{R}_7 = \begin{bmatrix} 1 & 1 & 1 & 1 & 1 & 1 & 1 \\ 1 & \cos \phi_7 & \cos 2\phi_7 & \cos 3\phi_7 & \cos 4\phi_7 & \cos 5\phi_7 & \cos 6\phi_7 \\ 1 & \cos 2\phi_7 & \cos 4\phi_7 & \cos 6\phi_7 & \cos 8\phi_7 & \cos 10\phi_7 & \cos 12\phi_7 \\ 1 & \cos 3\phi_7 & \cos 6\phi_7 & \cos 9\phi_7 & \cos 12\phi_7 & \cos 15\phi_7 & \cos 18\phi_7 \\ 0 & -\sin 4\phi_7 & -\sin 8\phi_7 & -\sin 12\phi_7 & -\sin 16\phi_7 & -\sin 20\phi_7 & -\sin 24\phi_7 \\ 0 & -\sin 5\phi_7 & -\sin 10\phi_7 & -\sin 15\phi_7 & -\sin 20\phi_7 & -\sin 25\phi_7 & -\sin 30\phi_7 \\ 0 & -\sin 6\phi_7 & -\sin 12\phi_7 & -\sin 18\phi_7 & -\sin 24\phi_7 & -\sin 30\phi_7 & -\sin 36\phi_7 \end{bmatrix} \tag{55}$$

Since $\phi_7 = 2\pi/7$ then $\cos 4\phi_7 = \cos 3\phi_7, \cos 5\phi_7 = \cos 2\phi_7, \cos 6\phi_7 = \cos \phi_7, \cos 8\phi_7 = \cos \phi_7, \cos 9\phi_7 = \cos 2\phi_7, \cos 10\phi_7 = \cos 3\phi_7, \cos 12\phi_7 = \cos 2\phi_7, \cos 15\phi_7 = \cos \phi_7, \cos 18\phi_7 = \cos 3\phi_7$ and $\sin 4\phi_7 = -\sin 3\phi_7, \sin 5\phi_7 = -\sin 2\phi_7, \sin 6\phi_7 = -\sin \phi_7, \sin 8\phi_7 = \sin \phi_7, \sin 10\phi_7 = \sin 3\phi_7, \sin 12\phi_7 = -\sin 2\phi_7, \sin 15\phi_7 = \sin \phi_7, \sin 16\phi_7 =$

$\sin 2\phi_7$, $\sin 18\phi_7 = -\sin 3\phi_7$, $\sin 20\phi_7 = -\sin\phi_7$, $\sin 24\phi_7 = \sin 3\phi_7$, $\sin 25\phi_7 = -\sin 3\phi_7$, $\sin 30\phi_7 = \sin 2\phi_7$, $\sin 36\phi_7 = \sin\phi_7$, so the $\mathbf{R}_7$ matrix will take the form

$$\mathbf{R}_7 = \begin{bmatrix} 1 & 1 & 1 & 1 & 1 & 1 & 1 \\ 1 & \cos\phi_7 & \cos 2\phi_7 & \cos 3\phi_7 & \cos 3\phi_7 & \cos 2\phi_7 & \cos\phi_7 \\ 1 & \cos 2\phi_7 & \cos 3\phi_7 & \cos\phi_7 & \cos\phi_7 & \cos 3\phi_7 & \cos 2\phi_7 \\ 1 & \cos 3\phi_7 & \cos\phi_7 & \cos 2\phi_7 & \cos 2\phi_7 & \cos\phi_7 & \cos 3\phi_7 \\ 0 & \sin 3\phi_7 & -\sin\phi_7 & \sin 2\phi_7 & -\sin 2\phi_7 & \sin\phi_7 & -\sin 3\phi_7 \\ 0 & \sin 2\phi_7 & -\sin 3\phi_7 & -\sin\phi_7 & \sin\phi_7 & \sin 3\phi_7 & -\sin 2\phi_7 \\ 0 & \sin\phi_7 & \sin 2\phi_7 & \sin 3\phi_7 & -\sin 3\phi_7 & -\sin 2\phi_7 & -\sin\phi_7 \end{bmatrix} \tag{56}$$

When we calculate the product of this matrix by the input vector $\mathbf{x}_7$ we obtain

$$\begin{bmatrix} y_0 \\ y_1 \\ y_2 \\ y_3 \\ y_4 \\ y_5 \\ y_6 \end{bmatrix} = \begin{bmatrix} x_0 + [(x_1 + x_6) + (x_2 + x_5) + (x_3 + x_4)] \\ x_0 + \cos\phi_7(x_1 + x_6) + \cos 2\phi_7(x_2 + x_5) + \cos 3\phi_7(x_3 + x_4) \\ x_0 + \cos 2\phi_7(x_1 + x_6) + \cos 3\phi_7(x_2 + x_5) + \cos\phi_7(x_3 + x_4) \\ x_0 + \cos 3\phi_7(x_1 + x_6) + \cos\phi_7(x_2 + x_5) + \cos 2\phi_7(x_3 + x_4) \\ \sin 3\phi_7(x_1 - x_6) - \sin\phi_7(x_2 - x_5) + \sin 2\phi_7(x_3 - x_4) \\ \sin 2\phi_7(x_1 - x_6) - \sin 3\phi_7(x_2 - x_5) - \sin\phi_7(x_3 - x_4) \\ \sin\phi_7(x_1 - x_6) + \sin 2\phi_7(x_2 - x_5) + \sin 3\phi_7(x_3 - x_4) \end{bmatrix} \tag{57}$$

To better understand the construction of the RDFT algorithm for $N = 7$, we will introduce the notations $t_1 = x_1 + x_6$, $t_2 = x_2 + x_5$, $t_3 = x_3 + x_4$, $t_4 = x_3 - x_4$, $t_5 = x_2 - x_5$, $t_6 = x_1 - x_6$, $t_0 = x_0 + t_1 + t_2 + t_3$ and consider the sub-blocks of the output vector $\mathbf{y}_7$. The first sub-block is

$$\begin{bmatrix} y_1 \\ y_2 \\ y_3 \end{bmatrix} = \begin{bmatrix} x_0 \\ x_0 \\ x_0 \end{bmatrix} + \begin{bmatrix} \cos\phi_7 & \cos 2\phi_7 & \cos 3\phi_7 \\ \cos 2\phi_7 & \cos 3\phi_7 & \cos\phi_7 \\ \cos 3\phi_7 & \cos\phi_7 & \cos 2\phi_7 \end{bmatrix} \begin{bmatrix} t_1 \\ t_2 \\ t_3 \end{bmatrix} =$$

$$= \begin{bmatrix} x_0 \\ x_0 \\ x_0 \end{bmatrix} + \begin{bmatrix} 1 & 1 & 1 & 0 \\ 1 & -1 & 0 & 1 \\ 1 & 0 & -1 & -1 \end{bmatrix} \begin{bmatrix} \hat{d}_1 & 0 & 0 & 0 \\ 0 & d_2 & 0 & 0 \\ 0 & 0 & d_3 & 0 \\ 0 & 0 & 0 & d_4 \end{bmatrix} \begin{bmatrix} 1 & 1 & 1 \\ 1 & 0 & -1 \\ 0 & 1 & -1 \\ -1 & 1 & 0 \end{bmatrix} \begin{bmatrix} t_1 \\ t_2 \\ t_3 \end{bmatrix} \tag{58}$$

where $\hat{d}_1 = \dfrac{1}{3}(\cos\phi_7 + \cos 2\phi_7 + \cos 3\phi_7)$, $d_2 = \dfrac{1}{3}(2\cos\phi_7 - \cos 2\phi_7 - \cos 3\phi_7)$, $d_3 = \dfrac{1}{3}(-\cos\phi_7 + 2\cos 2\phi_7 - \cos 3\phi_7)$, $d_4 = \dfrac{1}{3}(-\cos\phi_7 - \cos 2\phi_7 + 2\cos 3\phi_7)$.

It can also be written as

$$\begin{bmatrix} y_1 \\ y_2 \\ y_3 \end{bmatrix} = \begin{bmatrix} t_0 \\ t_0 \\ t_0 \end{bmatrix} + \begin{bmatrix} 1 & 1 & 1 & 0 \\ 1 & -1 & 0 & 1 \\ 1 & 0 & -1 & -1 \end{bmatrix} \begin{bmatrix} d_1 & 0 & 0 & 0 \\ 0 & d_2 & 0 & 0 \\ 0 & 0 & d_3 & 0 \\ 0 & 0 & 0 & d_4 \end{bmatrix} \begin{bmatrix} 1 & 1 & 1 \\ 1 & 0 & -1 \\ 0 & 1 & -1 \\ -1 & 1 & 0 \end{bmatrix} \begin{bmatrix} t_1 \\ t_2 \\ t_3 \end{bmatrix} \tag{59}$$

where $d_1 = \dfrac{1}{3}(\cos\phi_7 + \cos 2\phi_7 + \cos 3\phi_7) - 1$. The second sub-block is

$$\begin{bmatrix} y_4 \\ y_5 \\ y_6 \end{bmatrix} = \begin{bmatrix} \sin 2\phi_7 & -\sin\phi_7 & \sin 3\phi_7 \\ -\sin\phi_7 & -\sin 3\phi_7 & \sin 2\phi_7 \\ \sin 3\phi_7 & \sin 2\phi_7 & \sin\phi_7 \end{bmatrix} \begin{bmatrix} t_4 \\ t_5 \\ t_6 \end{bmatrix} =$$

$$= \begin{bmatrix} -1 & 0 & 1 & -1 \\ 1 & -1 & 0 & -1 \\ 1 & 1 & 1 & 0 \end{bmatrix} \begin{bmatrix} d_5 & 0 & 0 & 0 \\ 0 & d_6 & 0 & 0 \\ 0 & 0 & d_7 & 0 \\ 0 & 0 & 0 & d_8 \end{bmatrix} \begin{bmatrix} -1 & 1 & 1 \\ 1 & 0 & 1 \\ -1 & -1 & 0 \\ 0 & 1 & -1 \end{bmatrix} \begin{bmatrix} t_4 \\ t_5 \\ t_6 \end{bmatrix} \tag{60}$$

where $d_5 = \dfrac{1}{3}(\sin\phi_7 + \sin 2\phi_7 - \sin 3\phi_7)$, $d_6 = \dfrac{1}{3}(2\sin\phi_7 - \sin 2\phi_7 + \sin 3\phi_7)$, $d_7 = \dfrac{1}{3}(\sin\phi_7 - 2\sin 2\phi_7 - \sin 3\phi_7)$, $d_8 = \dfrac{1}{3}(\sin\phi_7 + \sin 2\phi_7 + 2\sin 3\phi_7)$.

Figure 5 shows a data-flow diagram corresponding to the calculation of the output vector $\mathbf{y}_7$.

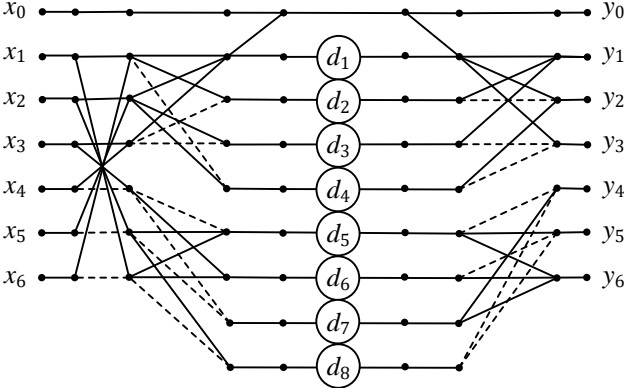

**Figure 5.** Data-flow diagram of the RDFT algorithm for $N = 7$.

When the vector $\mathbf{z}_7 = \mathbf{a}_7 + i\mathbf{b}_7$ of complex coefficients of DFT for the real input vector $\mathbf{x}_7$ is needed, it can be easily obtained from the output vector $\mathbf{y}_7$ of RDFT, according to (16)

$$
\mathbf{a}_7 = \begin{bmatrix} a_0 \\ a_1 \\ a_2 \\ a_3 \\ a_4 \\ a_5 \\ a_6 \end{bmatrix} = \begin{bmatrix} y_0 \\ y_1 \\ y_2 \\ y_3 \\ y_3 \\ y_2 \\ y_1 \end{bmatrix} \quad \mathbf{b}_7 = \begin{bmatrix} b_0 \\ b_1 \\ b_2 \\ b_3 \\ b_4 \\ b_5 \\ b_6 \end{bmatrix} = \begin{bmatrix} 0 \\ -y_6 \\ -y_5 \\ -y_4 \\ y_4 \\ y_5 \\ y_6 \end{bmatrix}
\tag{61}
$$

The algorithm of the RDFT for $N = 7$, presented in Figure 5, can be described by the following matrix–vector procedure, in which the matrix $\mathbf{R}_7$ has been factorized:

$$
\mathbf{y}_7 = \mathbf{C}_{7\times 9}\tilde{\mathbf{C}}_9\mathbf{D}_9\tilde{\mathbf{A}}_9\overline{\mathbf{A}}_{9\times 7}\hat{\mathbf{A}}_7\mathbf{x}_7
\tag{62}
$$

where

$$
\hat{\mathbf{A}}_7 = \begin{bmatrix} 1 & 0 & 0 & 0 & 0 & 0 & 0 \\ 0 & 1 & 0 & 0 & 0 & 0 & 1 \\ 0 & 0 & 1 & 0 & 0 & 1 & 0 \\ 0 & 0 & 0 & 1 & 1 & 0 & 0 \\ 0 & 0 & 0 & 1 & -1 & 0 & 0 \\ 0 & 0 & 1 & 0 & 0 & -1 & 0 \\ 0 & 1 & 0 & 0 & 0 & 0 & -1 \end{bmatrix} \quad \overline{\mathbf{A}}_{9\times 7} = \begin{bmatrix} 1 & 0 & 0 & 0 & 0 & 0 & 0 \\ 0 & 1 & 1 & 1 & 0 & 0 & 0 \\ 0 & 1 & 0 & -1 & 0 & 0 & 0 \\ 0 & 0 & 1 & -1 & 0 & 0 & 0 \\ 0 & -1 & 1 & 0 & 0 & 0 & 0 \\ 0 & 0 & 0 & 0 & -1 & 1 & 1 \\ 0 & 0 & 0 & 0 & 1 & 0 & 1 \\ 0 & 0 & 0 & 0 & -1 & -1 & 0 \\ 0 & 0 & 0 & 0 & 0 & 1 & -1 \end{bmatrix}
\tag{63}
$$

$$
\tilde{\mathbf{A}}_9 = \begin{bmatrix}
1 & 1 & 0 & 0 & 0 & 0 & 0 & 0 & 0 \\
0 & 1 & 0 & 0 & 0 & 0 & 0 & 0 & 0 \\
0 & 0 & 1 & 0 & 0 & 0 & 0 & 0 & 0 \\
0 & 0 & 0 & 1 & 0 & 0 & 0 & 0 & 0 \\
0 & 0 & 0 & 0 & 1 & 0 & 0 & 0 & 0 \\
0 & 0 & 0 & 0 & 0 & 1 & 0 & 0 & 0 \\
0 & 0 & 0 & 0 & 0 & 0 & 1 & 0 & 0 \\
0 & 0 & 0 & 0 & 0 & 0 & 0 & 1 & 0 \\
0 & 0 & 0 & 0 & 0 & 0 & 0 & 0 & 1
\end{bmatrix}
\quad
\mathbf{D}_9 = \begin{bmatrix}
1 & 0 & 0 & 0 & 0 & 0 & 0 & 0 & 0 \\
0 & d_1 & 0 & 0 & 0 & 0 & 0 & 0 & 0 \\
0 & 0 & d_2 & 0 & 0 & 0 & 0 & 0 & 0 \\
0 & 0 & 0 & d_3 & 0 & 0 & 0 & 0 & 0 \\
0 & 0 & 0 & 0 & d_4 & 0 & 0 & 0 & 0 \\
0 & 0 & 0 & 0 & 0 & d_5 & 0 & 0 & 0 \\
0 & 0 & 0 & 0 & 0 & 0 & d_6 & 0 & 0 \\
0 & 0 & 0 & 0 & 0 & 0 & 0 & d_7 & 0 \\
0 & 0 & 0 & 0 & 0 & 0 & 0 & 0 & d_8
\end{bmatrix}
\tag{64}
$$

$$
\tilde{\mathbf{C}}_9 = \begin{bmatrix}
1 & 0 & 0 & 0 & 0 & 0 & 0 & 0 & 0 \\
1 & 1 & 0 & 0 & 0 & 0 & 0 & 0 & 0 \\
0 & 0 & 1 & 0 & 0 & 0 & 0 & 0 & 0 \\
0 & 0 & 0 & 1 & 0 & 0 & 0 & 0 & 0 \\
0 & 0 & 0 & 0 & 1 & 0 & 0 & 0 & 0 \\
0 & 0 & 0 & 0 & 0 & 1 & 0 & 0 & 0 \\
0 & 0 & 0 & 0 & 0 & 0 & 1 & 0 & 0 \\
0 & 0 & 0 & 0 & 0 & 0 & 0 & 1 & 0 \\
0 & 0 & 0 & 0 & 0 & 0 & 0 & 0 & 1
\end{bmatrix}
\quad
\mathbf{C}_{7\times 9} = \begin{bmatrix}
1 & 0 & 0 & 0 & 0 & 0 & 0 & 0 & 0 \\
0 & 1 & 1 & 1 & 0 & 0 & 0 & 0 & 0 \\
0 & 1 & -1 & 0 & 1 & 0 & 0 & 0 & 0 \\
0 & 1 & 0 & -1 & -1 & 0 & 0 & 0 & 0 \\
0 & 0 & 0 & 0 & 0 & -1 & 0 & 1 & -1 \\
0 & 0 & 0 & 0 & 0 & 1 & -1 & 0 & -1 \\
0 & 0 & 0 & 0 & 0 & 1 & 1 & 1 & 0
\end{bmatrix}
\tag{65}
$$

According to this algorithm we need only 30 additions and 8 multiplications of real numbers to calculate the output vector $\mathbf{y}_7$.

## 8. RDFT Algorithm for $N = 8$

For $N = 8$, the Equation (17) will take the form

$$
\mathbf{y}_8 = \mathbf{R}_8 \mathbf{x}_8
\tag{66}
$$

where

$$
\mathbf{R}_8 = \begin{bmatrix}
1 & 1 & 1 & 1 & 1 & 1 & 1 & 1 \\
1 & \cos\phi_8 & \cos 2\phi_8 & \cos 3\phi_8 & \cos 4\phi_8 & \cos 5\phi_8 & \cos 6\phi_8 & \cos 7\phi_8 \\
1 & \cos 2\phi_8 & \cos 4\phi_8 & \cos 6\phi_8 & \cos 8\phi_8 & \cos 10\phi_8 & \cos 12\phi_8 & \cos 14\phi_8 \\
1 & \cos 3\phi_8 & \cos 6\phi_8 & \cos 9\phi_8 & \cos 12\phi_8 & \cos 15\phi_8 & \cos 18\phi_8 & \cos 21\phi_8 \\
1 & \cos 4\phi_8 & \cos 8\phi_8 & \cos 12\phi_8 & \cos 16\phi_8 & \cos 20\phi_8 & \cos 24\phi_8 & \cos 28\phi_8 \\
0 & -\sin 5\phi_8 & -\sin 10\phi_8 & -\sin 15\phi_8 & -\sin 20\phi_8 & -\sin 25\phi_8 & -\sin 30\phi_8 & -\sin 35\phi_8 \\
0 & -\sin 6\phi_8 & -\sin 12\phi_8 & -\sin 18\phi_8 & -\sin 24\phi_8 & -\sin 30\phi_8 & -\sin 36\phi_8 & -\sin 42\phi_8 \\
0 & -\sin 7\phi_8 & -\sin 14\phi_8 & -\sin 21\phi_8 & -\sin 28\phi_8 & -\sin 35\phi_8 & -\sin 42\phi_8 & -\sin 49\phi_8
\end{bmatrix}
\tag{67}
$$

Since $\phi_8 = 2\pi/8$ then $\cos 2\phi_0 = 0$, $\cos 3\phi_8 = -\cos\phi_8$, $\cos 4\phi_8 = -1$, $\cos 5\phi_8 = -\cos\phi_8$, $\cos 6\phi_8 = 0$, $\cos 7\phi_8 = \cos\phi_8$, $\cos 8\phi_8 = 1$, $\cos 9\phi_8 = \cos\phi_8$, $\cos 10\phi_8 = 0$, $\cos 12\phi_8 = -1$, $\cos 14\phi_8 = 0$, $\cos 15\phi_8 = \cos\phi_8$, $\cos 16\phi_8 = 1$, $\cos 18\phi_8 = 0$, $\cos 20\phi_8 = -1$, $\cos 21\phi_8 = -\cos\phi_8$, $\cos 24\phi_8 = 1$, $\cos 28\phi_8 = -1$, and $\sin 5\phi_8 = -\sin\phi_8$, $\sin 6\phi_8 = -1$, $\sin 7\phi_8 = -\sin\phi_8$, $\sin 10\phi_8 = 1$, $\sin 12\phi_8 = 0$, $\sin 14\phi_8 = -1$, $\sin 15\phi_8 = -\sin\phi_8$, $\sin 18\phi_8 = 1$, $\sin 20\phi_8 = 0$, $\sin 21\phi_8 = -\sin\phi_8$, $\sin 24\phi_8 = 0$, $\sin 25\phi_8 = \sin\phi_8$, $\sin 28\phi_8 = 0$, $\sin 30\phi_8 = -1$, $\sin 35\phi_8 = \sin\phi_8$, $\sin 36\phi_8 = 0$, $\sin 42\phi_8 = 1$, $\sin 49\phi_8 = \sin\phi_8$, so the $\mathbf{R}_8$ matrix will take the form

$$
\mathbf{R}_8 = \begin{bmatrix}
1 & 1 & 1 & 1 & 1 & 1 & 1 & 1 \\
1 & \cos\phi_8 & 0 & -\cos\phi_8 & -1 & -\cos\phi_8 & 0 & \cos\phi_8 \\
1 & 0 & -1 & 0 & 1 & 0 & -1 & 0 \\
1 & -\cos\phi_8 & 0 & \cos\phi_8 & -1 & \cos\phi_8 & 0 & -\cos\phi_8 \\
1 & -1 & 1 & -1 & 1 & -1 & 1 & -1 \\
0 & \sin\phi_8 & -1 & \sin\phi_8 & 0 & -\sin\phi_8 & 1 & -\sin\phi_8 \\
0 & 1 & 0 & -1 & 0 & 1 & 0 & -1 \\
0 & \sin\phi_8 & 1 & \sin\phi_8 & 0 & -\sin\phi_8 & -1 & -\sin\phi_8
\end{bmatrix}
\tag{68}
$$

When we calculate the product of this matrix by the input vector $\mathbf{x}_8$ we obtain

$$\begin{bmatrix} y_0 \\ y_1 \\ y_2 \\ y_3 \\ y_4 \\ y_5 \\ y_6 \\ y_7 \end{bmatrix} = \begin{bmatrix} [(x_0 + x_4) + (x_2 + x_6)] + [(x_1 + x_7) + (x_3 + x_5)] \\ (x_0 - x_4) + \cos\phi_8[(x_1 + x_7) - (x_3 + x_5)] \\ (x_0 + x_4) - (x_2 + x_6) \\ (x_0 - x_4) + \cos\phi_8[-(x_1 + x_7) + (x_3 + x_5)] \\ [(x_0 + x_4) + (x_2 + x_6)] - [(x_1 + x_7) + (x_3 + x_5)] \\ -(x_2 - x_6) + \sin\phi_8[(x_1 - x_7) - (x_3 - x_5)] \\ (x_1 - x_7) - (x_3 - x_5) \\ (x_2 - x_6) + \sin\phi_8[(x_1 - x_7) + (x_3 - x_5)] \end{bmatrix} \tag{69}$$

Figure 6 shows a data-flow diagram corresponding to the calculation of the output vector $\mathbf{y}_8$, where $d_3 = \cos\phi_8$ and $d_5 = \sin\phi_8$.

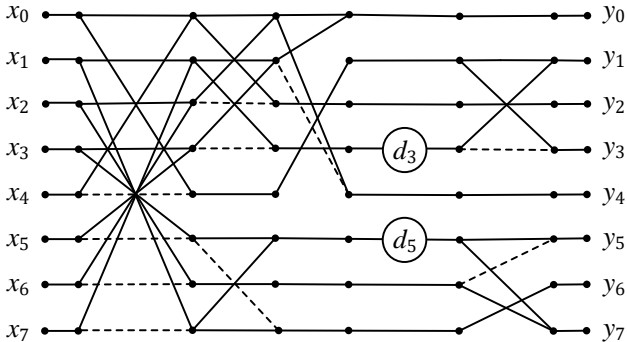

**Figure 6.** Data flow diagram of the RDFT algorithm for $N = 8$.

When the vector $\mathbf{z}_8 = \mathbf{a}_8 + i\mathbf{b}_8$ of complex coefficients of DFT for the real input vector $\mathbf{x}_8$ is needed, it can be easily obtained from the output vector $\mathbf{y}_8$ of RDFT, according to (15)

$$\mathbf{a}_8 = \begin{bmatrix} a_0 \\ a_1 \\ a_2 \\ a_3 \\ a_4 \\ a_5 \\ a_6 \\ a_7 \end{bmatrix} = \begin{bmatrix} y_0 \\ y_1 \\ y_2 \\ y_3 \\ y_4 \\ y_3 \\ y_2 \\ y_1 \end{bmatrix} \qquad \mathbf{b}_8 = \begin{bmatrix} b_0 \\ b_1 \\ b_2 \\ b_3 \\ b_4 \\ b_5 \\ b_6 \\ b_7 \end{bmatrix} = \begin{bmatrix} 0 \\ -y_7 \\ -y_6 \\ -y_5 \\ 0 \\ y_5 \\ y_6 \\ y_7 \end{bmatrix} \tag{70}$$

The algorithm of the RDFT for $N = 8$, presented in Figure 6, can be described by the following matrix–vector procedure, in which the matrix $\mathbf{R}_8$ has been factorized:

$$\mathbf{y}_8 = \mathbf{C}_8 \mathbf{D}_8 \tilde{\mathbf{A}}_8 \overline{\mathbf{A}}_8 \hat{\mathbf{A}}_8 \mathbf{x}_8 \tag{71}$$

where

$$\hat{\mathbf{A}}_8 = \begin{bmatrix} 1 & 0 & 0 & 0 & 1 & 0 & 0 & 0 \\ 0 & 1 & 0 & 0 & 0 & 0 & 0 & 1 \\ 0 & 0 & 1 & 0 & 0 & 0 & 1 & 0 \\ 0 & 0 & 0 & 1 & 0 & 1 & 0 & 0 \\ 1 & 0 & 0 & 0 & -1 & 0 & 0 & 0 \\ 0 & 0 & 0 & 1 & 0 & -1 & 0 & 0 \\ 0 & 0 & 1 & 0 & 0 & 0 & -1 & 0 \\ 0 & 1 & 0 & 0 & 0 & 0 & 0 & -1 \end{bmatrix} \quad \overline{\mathbf{A}}_8 = \begin{bmatrix} 1 & 0 & 1 & 0 & 0 & 0 & 0 & 0 \\ 0 & 1 & 0 & 1 & 0 & 0 & 0 & 0 \\ 1 & 0 & -1 & 0 & 0 & 0 & 0 & 0 \\ 0 & 1 & 0 & -1 & 0 & 0 & 0 & 0 \\ 0 & 0 & 0 & 0 & 1 & 0 & 0 & 0 \\ 0 & 0 & 0 & 0 & 0 & 1 & 0 & 1 \\ 0 & 0 & 0 & 0 & 0 & 0 & 1 & 0 \\ 0 & 0 & 0 & 0 & 0 & -1 & 0 & 1 \end{bmatrix} \tag{72}$$

$$\tilde{\mathbf{A}}_8 = \begin{bmatrix} 1 & 1 & 0 & 0 & 0 & 0 & 0 & 0 \\ 0 & 0 & 0 & 0 & 1 & 0 & 0 & 0 \\ 0 & 0 & 1 & 0 & 0 & 0 & 0 & 0 \\ 0 & 0 & 0 & 1 & 0 & 0 & 0 & 0 \\ 1 & -1 & 0 & 0 & 0 & 0 & 0 & 0 \\ 0 & 0 & 0 & 0 & 0 & 1 & 0 & 0 \\ 0 & 0 & 0 & 0 & 0 & 0 & 1 & 0 \\ 0 & 0 & 0 & 0 & 0 & 0 & 0 & 1 \end{bmatrix} \quad \mathbf{D}_8 = \begin{bmatrix} 1 & 0 & 0 & 0 & 0 & 0 & 0 & 0 \\ 0 & 1 & 0 & 0 & 0 & 0 & 0 & 0 \\ 0 & 0 & 1 & 0 & 0 & 0 & 0 & 0 \\ 0 & 0 & 0 & d_3 & 0 & 0 & 0 & 0 \\ 0 & 0 & 0 & 0 & 1 & 0 & 0 & 0 \\ 0 & 0 & 0 & 0 & 0 & d_5 & 0 & 0 \\ 0 & 0 & 0 & 0 & 0 & 0 & 1 & 0 \\ 0 & 0 & 0 & 0 & 0 & 0 & 0 & 1 \end{bmatrix} \tag{73}$$

$$\mathbf{C}_8 = \begin{bmatrix} 1 & 0 & 0 & 0 & 0 & 0 & 0 & 0 \\ 0 & 1 & 0 & 1 & 0 & 0 & 0 & 0 \\ 0 & 0 & 1 & 0 & 0 & 0 & 0 & 0 \\ 0 & 1 & 0 & -1 & 0 & 0 & 0 & 0 \\ 0 & 0 & 0 & 0 & 1 & 0 & 0 & 0 \\ 0 & 0 & 0 & 0 & 0 & 1 & -1 & 0 \\ 0 & 0 & 0 & 0 & 0 & 0 & 0 & 1 \\ 0 & 0 & 0 & 0 & 0 & 1 & 1 & 0 \end{bmatrix} \tag{74}$$

According to this algorithm we need only 20 additions and 2 multiplications of real numbers to calculate the output vector $\mathbf{y}_8$.

## 9. RDFT Algorithm for $N = 9$

For $N = 9$, the Equation (17) will take the form

$$\mathbf{y}_9 = \mathbf{R}_9 \mathbf{x}_9 \tag{75}$$

where

$$\mathbf{R}_9 = \begin{bmatrix} 1 & \mathbf{1}_{1\times4} & \mathbf{1}_{1\times4} \\ \mathbf{1}_{4\times1} & \mathbf{A}_4 & \mathbf{B}_4 \\ \mathbf{0}_{4\times1} & \mathbf{C}_4 & \mathbf{D}_4 \end{bmatrix} \tag{76}$$

and $\mathbf{1}_{n\times k}$, $\mathbf{0}_{n\times k}$ are $n$ by $k$ submatrices with all entries equal to 1 or 0, respectively. After applying the reduction formulas and taking advantage of the fact that $\cos 3\phi_9 = \cos(6\pi/9) = -0.5$, the component submatrices can be written in the following forms:

$$\mathbf{A}_4 = \begin{bmatrix} \cos\phi_9 & \cos 2\phi_9 & -0.5 & \cos 4\phi_9 \\ \cos 2\phi_9 & \cos 4\phi_9 & -0.5 & \cos\phi_9 \\ -0.5 & -0.5 & 1 & -0.5 \\ \cos 4\phi_9 & \cos\phi_9 & -0.5 & \cos 2\phi_9 \end{bmatrix} \tag{77}$$

$$\mathbf{B}_4 = \begin{bmatrix} \cos 4\phi_9 & -0.5 & \cos 2\phi_9 & \cos\phi_9 \\ \cos\phi_9 & -0.5 & \cos 4\phi_9 & \cos 2\phi_9 \\ -0.5 & 1 & -0.5 & -0.5 \\ \cos 2\phi_9 & -0.5 & \cos\phi_9 & \cos 4\phi_9 \end{bmatrix} \tag{78}$$

$$\mathbf{C}_4 = \begin{bmatrix} \sin 4\phi_9 & -\sin\phi_9 & \sin 3\phi_9 & -\sin 2\phi_9 \\ \sin 3\phi_9 & -\sin 3\phi_9 & 0 & \sin 3\phi_9 \\ \sin 2\phi_9 & \sin 4\phi_9 & -\sin 3\phi_9 & -\sin\phi_9 \\ \sin\phi_9 & \sin 2\phi_9 & \sin 3\phi_9 & \sin 4\phi_9 \end{bmatrix} \tag{79}$$

$$\mathbf{D}_4 = \begin{bmatrix} \sin 2\phi_9 & -\sin 3\phi_9 & \sin\phi_9 & -\sin 4\phi_9 \\ -\sin 3\phi_9 & 0 & \sin 3\phi_9 & -\sin 3\phi_9 \\ \sin\phi_9 & \sin 3\phi_9 & -\sin 4\phi_9 & -\sin 2\phi_9 \\ -\sin 4\phi_9 & -\sin 3\phi_9 & -\sin 2\phi_9 & -\sin\phi_9 \end{bmatrix} \tag{80}$$

It is easy to see that the matrix $\mathbf{B}_4$ can be obtained from the matrix $\mathbf{A}_4$ by reversing the order of its columns, and the matrix $\mathbf{D}_4$ is the opposite matrix to the matrix obtained from

$C_4$ by reversing the order of its columns. When we calculate the product of the matrix $\mathbf{R}_9$ by the input vector $\mathbf{x}_9$ we obtain

$$
\begin{bmatrix} y_0 \\ y_1 \\ y_2 \\ y_3 \\ y_4 \\ y_5 \\ y_6 \\ y_7 \\ y_8 \end{bmatrix} = \begin{bmatrix} [x_0 + (x_3+x_6)] + [(x_1+x_8) + (x_2+x_7) + (x_4+x_5)] \\ [x_0 - 0.5(x_3+x_6)] + \cos\phi_9[(x_1+x_8) + \cos 2\phi_9[(x_2+x_7) + \cos 4\phi_9[(x_4+x_5) \\ [x_0 - 0.5(x_3+x_6)] + \cos 2\phi_9[(x_1+x_8) + \cos 4\phi_9[(x_2+x_7) + \cos \phi_9[(x_4+x_5) \\ [x_0 + (x_3+x_6)] - 0.5[(x_1+x_8) + (x_2+x_7) + (x_4+x_5)] \\ [x_0 - 0.5(x_3+x_6)] + \cos 4\phi_9[(x_1+x_8) + \cos \phi_9[(x_2+x_7) + \cos 2\phi_9[(x_4+x_5) \\ \sin 3\phi_9(x_3 - x_6)] - \sin 2\phi_9[(x_4 - x_5) - \sin \phi_9[(x_2 - x_7) + \sin 4\phi_9[(x_1 - x_8) \\ \sin 3\phi_9[(x_4 - x_5) - (x_2 - x_7) + (x_1 - x_8)] \\ -\sin 3\phi_9(x_3 - x_6)] - \sin \phi_9[(x_4 - x_5) + \sin 4\phi_9[(x_2 - x_7) + \sin 2\phi_9[(x_1 - x_8) \\ \sin 3\phi_9(x_3 - x_6)] + \sin 4\phi_9[(x_4 - x_5) + \sin 2\phi_9[(x_2 - x_7) + \sin \phi_9[(x_1 - x_8) \end{bmatrix} \quad (81)
$$

To better understand the construction of the RDFT algorithm for $N = 9$, we will introduce the notations $t_1 = x_1 + x_8$, $t_2 = x_2 + x_7$, $t_3 = x_3 + x_6$, $t_4 = x_4 + x_5$, $t_5 = x_4 - x_5$, $t_6 = x_3 - x_6$, $t_7 = x_2 - x_7$, $t_8 = x_1 - x_8$ and consider the sub-blocks of the output vector $\mathbf{y}_8$. The first sub-block is

$$
\begin{bmatrix} y_1 \\ y_2 \\ y_4 \end{bmatrix} = \begin{bmatrix} x_0 - 0.5t_3 \\ x_0 - 0.5t_3 \\ x_0 - 0.5t_3 \end{bmatrix} + \begin{bmatrix} \cos\phi_9 & \cos 2\phi_9 & \cos 4\phi_9 \\ \cos 2\phi_9 & \cos 4\phi_9 & \cos \phi_9 \\ \cos 4\phi_9 & \cos \phi_9 & \cos 2\phi_9 \end{bmatrix} \begin{bmatrix} t_1 \\ t_2 \\ t_4 \end{bmatrix} =
$$

$$
= \begin{bmatrix} x_0 - 0.5t_3 \\ x_0 - 0.5t_3 \\ x_0 - 0.5t_3 \end{bmatrix} + \begin{bmatrix} 1 & 1 & 1 & 0 \\ 1 & -1 & 0 & 1 \\ 1 & 0 & -1 & -1 \end{bmatrix} \begin{bmatrix} d_5 & 0 & 0 & 0 \\ 0 & d_6 & 0 & 0 \\ 0 & 0 & d_7 & 0 \\ 0 & 0 & 0 & d_8 \end{bmatrix} \begin{bmatrix} 1 & 1 & 1 \\ 1 & 0 & -1 \\ 0 & 1 & -1 \\ -1 & 1 & 0 \end{bmatrix} \begin{bmatrix} t_1 \\ t_2 \\ t_4 \end{bmatrix} \quad (82)
$$

where $d_5 = \dfrac{1}{3}(\cos\phi_9 + \cos 2\phi_9 + \cos 4\phi_9)$, $d_6 = \dfrac{1}{3}(2\cos\phi_9 - \cos 2\phi_9 - \cos 4\phi_9)$, $d_7 = \dfrac{1}{3}(-\cos\phi_9 + 2\cos 2\phi_9 - \cos 4\phi_9)$, $d_8 = \dfrac{1}{3}(-\cos\phi_9 - \cos 2\phi_9 + 2\cos 3\phi_9)$.

The second sub-block is

$$
\begin{bmatrix} y_5 \\ y_7 \\ y_8 \end{bmatrix} = \begin{bmatrix} t_4 \sin 3\phi_9 \\ -t_4 \sin 3\phi_9 \\ t_4 \sin 3\phi_9 \end{bmatrix} + \begin{bmatrix} -\sin 2\phi_9 & -\sin \phi_9 & \sin 4\phi_9 \\ -\sin \phi_9 & \sin 4\phi_9 & \sin 2\phi_9 \\ \sin 4\phi_9 & \sin 2\phi_9 & \sin \phi_9 \end{bmatrix} \begin{bmatrix} t_5 \\ t_7 \\ t_8 \end{bmatrix} =
$$

$$
= \begin{bmatrix} t_4 \sin 3\phi_9 \\ -t_4 \sin 3\phi_9 \\ t_4 \sin 3\phi_9 \end{bmatrix} + \begin{bmatrix} 1 & 1 & 1 & 0 \\ -1 & 1 & 0 & -1 \\ 1 & 0 & -1 & -1 \end{bmatrix} \begin{bmatrix} d_9 & 0 & 0 & 0 \\ 0 & d_{10} & 0 & 0 \\ 0 & 0 & d_{11} & 0 \\ 0 & 0 & 0 & d_{12} \end{bmatrix} \begin{bmatrix} 1 & -1 & 1 \\ 1 & 0 & -1 \\ 0 & -1 & -1 \\ -1 & -1 & 0 \end{bmatrix} \begin{bmatrix} t_5 \\ t_7 \\ t_8 \end{bmatrix} \quad (83)
$$

where $d_9 = \dfrac{1}{3}(\sin\phi_9 - \sin 2\phi_9 + \sin 4\phi_9)$, $d_{10} = \dfrac{1}{3}(-\sin\phi_9 - 2\sin 2\phi_9 - \sin 4\phi_9)$, $d_{11} = \dfrac{1}{3}(2\sin\phi_9 + \sin 2\phi_9 - \sin 4\phi_9)$, $d_{12} = \dfrac{1}{3}(-\sin\phi_9 + \sin 2\phi_9 + 2\sin 4\phi_9)$.

Figure 7 shows a data flow diagram corresponding to the calculation of the output vector $\mathbf{y}_9$, where $d_{13} = d_{14} = \sin 3\phi_9$.

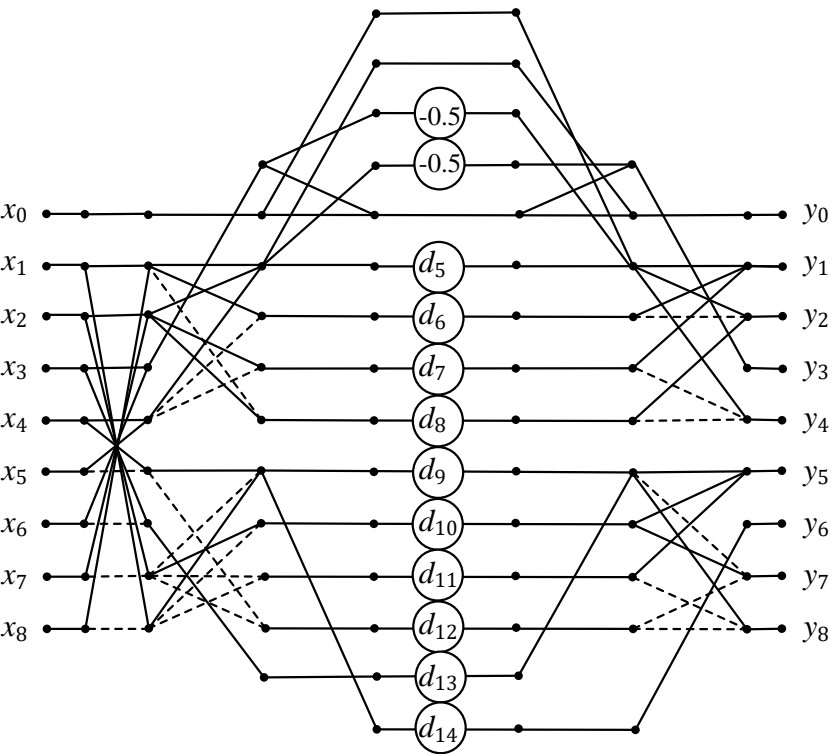

**Figure 7.** Data–flow diagram of the RDFT algorithm for $N = 9$.

When the vector $\mathbf{z}_9 = \mathbf{a}_a + i\mathbf{b}_9$ of complex coefficients of DFT for the real input vector $\mathbf{x}_9$ is needed, it can be easily obtained from the output vector $\mathbf{y}_9$ of RDFT, according to (16)

$$
\mathbf{a}_9 = \begin{bmatrix} a_0 \\ a_1 \\ a_2 \\ a_3 \\ a_4 \\ a_5 \\ a_6 \\ a_7 \\ a_8 \end{bmatrix} = \begin{bmatrix} y_0 \\ y_1 \\ y_2 \\ y_3 \\ y_4 \\ y_4 \\ y_3 \\ y_2 \\ y_1 \end{bmatrix} \qquad \mathbf{b}_7 = \begin{bmatrix} b_0 \\ b_1 \\ b_2 \\ b_3 \\ b_4 \\ b_5 \\ b_6 \\ b_7 \\ b_8 \end{bmatrix} = \begin{bmatrix} 0 \\ -y_8 \\ -y_7 \\ -y_6 \\ -y_5 \\ y_5 \\ y_6 \\ y_7 \\ y_8 \end{bmatrix} \tag{84}
$$

The algorithm of the RDFT for $N = 9$, presented in Figure 7, can be described by the following matrix–vector procedure, in which the matrix $\mathbf{R}_9$ has been factorized:

$$
\mathbf{y}_9 = \mathbf{C}_{9 \times 11} \tilde{\mathbf{C}}_{11 \times 15} \mathbf{D}_{15} \tilde{\mathbf{A}}_{15 \times 11} \overline{\mathbf{A}}_{11 \times 9} \hat{\mathbf{A}}_9 \mathbf{x}_9 \tag{85}
$$

where

$$
\hat{\mathbf{A}}_9 = \begin{bmatrix}
1 & 0 & 0 & 0 & 0 & 0 & 0 & 0 & 0 \\
0 & 1 & 0 & 0 & 0 & 0 & 0 & 0 & 1 \\
0 & 0 & 1 & 0 & 0 & 0 & 0 & 1 & 0 \\
0 & 0 & 0 & 1 & 0 & 0 & 1 & 0 & 0 \\
0 & 0 & 0 & 0 & 1 & 1 & 0 & 0 & 0 \\
0 & 0 & 0 & 0 & 1 & -1 & 0 & 0 & 0 \\
0 & 0 & 0 & 1 & 0 & 0 & -1 & 0 & 0 \\
0 & 0 & 1 & 0 & 0 & 0 & 0 & -1 & 0 \\
0 & 1 & 0 & 0 & 0 & 0 & 0 & 0 & -1
\end{bmatrix} \tag{86}
$$

$$
\overline{\mathbf{A}}_{11\times9} =
\begin{bmatrix}
0 & 0 & 0 & 1 & 0 & 0 & 0 & 0 & 0 \\
1 & 0 & 0 & 0 & 0 & 0 & 0 & 0 & 0 \\
0 & 1 & 1 & 0 & 1 & 0 & 0 & 0 & 0 \\
0 & 1 & 0 & 0 & -1 & 0 & 0 & 0 & 0 \\
0 & 0 & 1 & 0 & -1 & 0 & 0 & 0 & 0 \\
0 & -1 & 1 & 0 & 0 & 0 & 0 & 0 & 0 \\
0 & 0 & 0 & 0 & 0 & 1 & 0 & -1 & 1 \\
0 & 0 & 0 & 0 & 0 & 1 & 0 & 0 & -1 \\
0 & 0 & 0 & 0 & 0 & 0 & 0 & -1 & -1 \\
0 & 0 & 0 & 0 & 0 & -1 & 0 & -1 & 0 \\
0 & 0 & 0 & 0 & 0 & 0 & 1 & 0 & 0
\end{bmatrix}
\tag{87}
$$

$$
\tilde{\mathbf{A}}_{15\times11} =
\begin{bmatrix}
0 & 1 & 0 & 0 & 0 & 0 & 0 & 0 & 0 & 0 & 0 \\
0 & 0 & 1 & 0 & 0 & 0 & 0 & 0 & 0 & 0 & 0 \\
1 & 0 & 0 & 0 & 0 & 0 & 0 & 0 & 0 & 0 & 0 \\
0 & 0 & 1 & 0 & 0 & 0 & 0 & 0 & 0 & 0 & 0 \\
1 & 1 & 0 & 0 & 0 & 0 & 0 & 0 & 0 & 0 & 0 \\
0 & 0 & 1 & 0 & 0 & 0 & 0 & 0 & 0 & 0 & 0 \\
0 & 0 & 0 & 1 & 0 & 0 & 0 & 0 & 0 & 0 & 0 \\
0 & 0 & 0 & 0 & 1 & 0 & 0 & 0 & 0 & 0 & 0 \\
0 & 0 & 0 & 0 & 0 & 1 & 0 & 0 & 0 & 0 & 0 \\
0 & 0 & 0 & 0 & 0 & 0 & 1 & 0 & 0 & 0 & 0 \\
0 & 0 & 0 & 0 & 0 & 0 & 0 & 1 & 0 & 0 & 0 \\
0 & 0 & 0 & 0 & 0 & 0 & 0 & 0 & 1 & 0 & 0 \\
0 & 0 & 0 & 0 & 0 & 0 & 0 & 0 & 0 & 1 & 0 \\
0 & 0 & 0 & 0 & 0 & 0 & 0 & 0 & 0 & 0 & 1 \\
0 & 0 & 0 & 0 & 0 & 0 & 1 & 0 & 0 & 0 & 0
\end{bmatrix}
\tag{88}
$$

$$
\mathbf{D}_{15} =
\begin{bmatrix}
1 & 0 & 0 & 0 & 0 & 0 & 0 & 0 & 0 & 0 & 0 & 0 & 0 & 0 & 0 \\
0 & 1 & 0 & 0 & 0 & 0 & 0 & 0 & 0 & 0 & 0 & 0 & 0 & 0 & 0 \\
0 & 0 & -0.5 & 0 & 0 & 0 & 0 & 0 & 0 & 0 & 0 & 0 & 0 & 0 & 0 \\
0 & 0 & 0 & -0.5 & 0 & 0 & 0 & 0 & 0 & 0 & 0 & 0 & 0 & 0 & 0 \\
0 & 0 & 0 & 0 & 1 & 0 & 0 & 0 & 0 & 0 & 0 & 0 & 0 & 0 & 0 \\
0 & 0 & 0 & 0 & 0 & d_5 & 0 & 0 & 0 & 0 & 0 & 0 & 0 & 0 & 0 \\
0 & 0 & 0 & 0 & 0 & 0 & d_6 & 0 & 0 & 0 & 0 & 0 & 0 & 0 & 0 \\
0 & 0 & 0 & 0 & 0 & 0 & 0 & d_7 & 0 & 0 & 0 & 0 & 0 & 0 & 0 \\
0 & 0 & 0 & 0 & 0 & 0 & 0 & 0 & d_8 & 0 & 0 & 0 & 0 & 0 & 0 \\
0 & 0 & 0 & 0 & 0 & 0 & 0 & 0 & 0 & d_9 & 0 & 0 & 0 & 0 & 0 \\
0 & 0 & 0 & 0 & 0 & 0 & 0 & 0 & 0 & 0 & d_{10} & 0 & 0 & 0 & 0 \\
0 & 0 & 0 & 0 & 0 & 0 & 0 & 0 & 0 & 0 & 0 & d_{11} & 0 & 0 & 0 \\
0 & 0 & 0 & 0 & 0 & 0 & 0 & 0 & 0 & 0 & 0 & 0 & d_{12} & 0 & 0 \\
0 & 0 & 0 & 0 & 0 & 0 & 0 & 0 & 0 & 0 & 0 & 0 & 0 & d_{13} & 0 \\
0 & 0 & 0 & 0 & 0 & 0 & 0 & 0 & 0 & 0 & 0 & 0 & 0 & 0 & d_{14}
\end{bmatrix}
\tag{89}
$$

$$
\tilde{\mathbf{C}}_{11\times15} =
\begin{bmatrix}
0 & 0 & 0 & 1 & 1 & 0 & 0 & 0 & 0 & 0 & 0 & 0 & 0 & 0 & 0 \\
1 & 0 & 0 & 0 & 1 & 0 & 0 & 0 & 0 & 0 & 0 & 0 & 0 & 0 & 0 \\
1 & 0 & 1 & 0 & 0 & 1 & 0 & 0 & 0 & 0 & 0 & 0 & 0 & 0 & 0 \\
0 & 0 & 0 & 0 & 0 & 0 & 1 & 0 & 0 & 0 & 0 & 0 & 0 & 0 & 0 \\
0 & 0 & 0 & 0 & 0 & 0 & 0 & 1 & 0 & 0 & 0 & 0 & 0 & 0 & 0 \\
0 & 0 & 0 & 0 & 0 & 0 & 0 & 0 & 1 & 0 & 0 & 0 & 0 & 0 & 0 \\
0 & 0 & 0 & 0 & 0 & 0 & 0 & 0 & 0 & 1 & 0 & 0 & 0 & 1 & 0 \\
0 & 0 & 0 & 0 & 0 & 0 & 0 & 0 & 0 & 0 & 1 & 0 & 0 & 0 & 0 \\
0 & 0 & 0 & 0 & 0 & 0 & 0 & 0 & 0 & 0 & 0 & 1 & 0 & 0 & 0 \\
0 & 0 & 0 & 0 & 0 & 0 & 0 & 0 & 0 & 0 & 0 & 0 & 1 & 0 & 0 \\
0 & 0 & 0 & 0 & 0 & 0 & 0 & 0 & 0 & 0 & 0 & 0 & 0 & 0 & 1
\end{bmatrix}
\tag{90}
$$

$$\mathbf{C}_{9 \times 11} = \begin{bmatrix} 0 & 1 & 0 & 0 & 0 & 0 & 0 & 0 & 0 & 0 & 0 \\ 0 & 0 & 1 & 1 & 1 & 0 & 0 & 0 & 0 & 0 & 0 \\ 0 & 0 & 1 & -1 & 0 & 1 & 0 & 0 & 0 & 0 & 0 \\ 1 & 0 & 0 & 0 & 0 & 0 & 0 & 0 & 0 & 0 & 0 \\ 0 & 0 & 1 & 0 & -1 & -1 & 0 & 0 & 0 & 0 & 0 \\ 0 & 0 & 0 & 0 & 0 & 0 & 1 & 1 & 1 & 0 & 0 \\ 0 & 0 & 0 & 0 & 0 & 0 & 0 & 0 & 0 & 0 & 1 \\ 0 & 0 & 0 & 0 & 0 & 0 & -1 & 1 & 0 & -1 & 0 \\ 0 & 0 & 0 & 0 & 0 & 0 & 1 & 0 & -1 & -1 & 0 \end{bmatrix} \tag{91}$$

According to this algorithm, we need only 36 additions and 10 multiplications (and 2 shifts) of real numbers to calculate the output vector $\mathbf{y}_9$.

## 10. Discussion

Table 1 compares the number of multiplications ($\times$) and additions ($+$) of real numbers necessary to determine the DFTs for real data vectors according to the proposed algorithms and Winograd's algorithms designed specifically for small data lengths $N$. We compare our solutions with the corresponding Winograd's algorithms because although these solutions are quite old they are the best in terms of multiplicative complexity. It should be remembered that the hardware multiplier is a very resource-intensive unit. The multiplier is the most resource-intensive and energy-consuming arithmetic unit, occupying a large area of the chip and dissipating a lot of power. Therefore, the use of complex and resource-intensive FPGAs containing a large number of multipliers without a special need is impractical. They use more power, take up more PCB space, and generate more heat than simpler chips. Thus, using less sophisticated chips and lower thermal management overheads translates into reduced processor size, weight, power consumption, and cost, as well as increased reliability as an added benefit.

The advantage of the presented Winograd-type algorithms in comparison with the Cooley–Tukey algorithms is that the critical path in the graph of any of the obtained algorithms contains only one multiplication. If there is more than one multiplication in the critical path of the algorithm, then this will create additional problems for the implementation of computations. As a result of multiplying two $n$-bit operands, a $2n$-bit product is obtained. The need for repeated multiplication requires an additional amount of manipulations with the operands and therefore requires more time and effort than when we are dealing with only a single multiplication.

**Table 1.** Comparison of the number of arithmetic operations for the proposed algorithms and Winograd's algorithms described in [1].

| $N$ | Proposed Solution | | Winograd's Small-Lengths DFTs | |
|---|---|---|---|---|
| | $\times$ | $+$ | $\times$ | $+$ |
| 3 | 2 | 4 | 2 | 6 |
| 4 | 0 | 6 | 0 | 8 |
| 5 | 5 | 13 | 5 | 17 |
| 6 | 4 | 14 | - | - |
| 7 | 8 | 30 | 8 | 36 |
| 8 | 2 | 20 | 2 | 26 |
| 9 | 10 | 36 | 10 | 44 |

It is easy to observe that the numbers of necessary multiplications are the same for the proposed algorithms as in Winograd's algorithms, but the number of needed additions is smaller in the case of the solutions presented in the article. It must be said that the difference in the number of addition operations for the compared solutions is minimal and is not the main trump card of our paper. Our main goal was to reveal those aspects and features of the organization of calculations of small-size real-valued DFTs that were not disclosed in the available literature.

## 11. Conclusions

The paper presents a complete set of algorithms for real short-length RDFTs for $N$ from 3 to 9. The corresponding signal flow graphs are also presented. The structure of each such graph, if necessary, can be directly mapped to the VLSI structure. The described algorithms are written in matrix–vector notation, where RDFT matrices are factorized and the factors are sparse matrices. This factorization reduces the number of arithmetic operations. Although the presented algorithms do not have a repeating structure for different lengths of input vectors, in some cases they may be more applicable and convenient in terms of implementation. One way or another, the solutions proposed by us are original and may be helpful. It should be emphasized that the diversity of existing approaches to the optimization of calculations cannot serve as an argument for stopping the search for new solutions, which may be more effective from the point of view of previously neglected criteria. Therefore, any rational approach to solving the current problem has the right to exist because each new look and each new solution of a known issue, which was previously solved by another method, stimulates the development of theory and practice, expands and deepens our knowledge in the relevant field of science or technology and, at least from this point of view, is helpful.

**Author Contributions:** Conceptualization, A.C.; methodology, A.C. and D.M.-M.; formal analysis, D.M.-M.; writing—original draft preparation, D.M.-M.; writing—review and editing, D.M.-M.; visualization, D.M.-M.; supervision, A.C. All authors have read and agreed to the published version of the manuscript.

**Funding:** This research received no external funding.

**Institutional Review Board Statement:** Not applicable.

**Informed Consent Statement:** Not applicable.

**Data Availability Statement:** Not applicable.

**Conflicts of Interest:** The authors declare no conflict of interest.

## Abbreviations

The following abbreviations are used in this manuscript:

| | |
|---|---|
| DFT | discrete Fourier transform |
| FFT | fast Fourier transform |
| RDFT | real discrete Fourier transform |
| FPGA | field programmable gate array |
| PCB | printed circuit board |
| VLSI | very large-scale integration |

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
