# Peer review of "Some FFT Algorithms for Small-Length Real-Valued Sequences"

_applsci, doi:10.3390/app12094700_

Round 1

Reviewer 1 Report

Some new algorithms for a small-size real data FFT are presented. The topic is interesting but there are some observations that could be important:

-in the presentation of the state of the art it would be useful to present some other similar solutions for a small-size FFT and to present their advantages and drawbacks.

-a discussion of the results it would be necessary

-a comparison with existing solutions for small-size FFT algorithms it would be really useful

Reviewer 2 Report

The paper describes the fast algorithms for computing the DFT for real-valued sequences of length from 2 to 8. The author specifies the problem area as in the practical applications of digital signal processing, the dataset is mostly real-valued. The limitations of the manuscript are as follows:

(1) In your Introduction, you state that your research problem is not sufficiently addressed in the existing literature. I suggest you discuss how your finding compare and contrast to previous research in detail.

(2) I would encourage the authors to include a discussion section in their paper discussing how their algorithms can be verified and validated to meet the needs of digital signal processing applications in practical.

Reviewer 3 Report

The authors researched an FFT algorithm for so-called small-length real-valued sequences. They considered lengths from two (2) to eight (8). They intend to reduce arithmetical operations.  

This is an exciting research report. In my opinion, the research subject is actual and has the potential to be published. However, the manuscript is not finished, and in its current form, it can not be recommended for publication.

I have the following comments and suggestions:

  1. The novelty of the manuscript is not clear and emphasized enough. 
  2. It is not clear what are aims of the article? Is it to present "a complete imagination about the organization of the small-size real-valued DFT calculation process" (L47) or "minimizing the number of multiplications" (L220)?
  3. There is no discussion section. Please, compare your research with other research (such as FFT and DFT algorithms). Please, provide metrics to convince the potential reader.
  4. Since you applied to publish your work in Applied Science Journal, please, provide an example or potential application of the proposed algorithm.

Round 2

Reviewer 1 Report

The new ideas could be still better presented.

The discussion section is present now but it could be also improved and some comparison with similar short length implementations would be useful. I think is not enough only to compare with the Winograd's algorithm which is really very old.

Reviewer 2 Report

This paper proposes a fast algorithms for computing the discrete Fourier transform for real-valued sequences of length from 3 to 8. 

Reviewer 3 Report

The authors improved the manuscript to some degree. In my opinion, the subject is interesting and current, and it has the potential to be published. However, there is room for further improvement in the article as follows:

  1. Why have you used an algorithm for sample lengths from three (3) to eight (8) and not from three to ten (10) or eleven (11) since there is no repeating structure? 
  2. For example, there is no discussion about what is better for n=8 (n, number of samples) to use the algorithm to perform one calculation (choose N=8) or choose N=4 and perform "two" calculations (due to resolution problem)?
  3. What are the potential limitations of a proposed algorithm?
  4. How an algorithm behaves on the edges? Furthermore, what about the robustness of the algorithm?

Round 3

Reviewer 1 Report

The paper have been improved but the discussion section still can be improved.

Author Response

Following the recommendations of a respected Reviewer, we supplemented the text of the manuscript. So, we thank the respected Reviewer for his efforts and help in improving the quality of our manuscript.

Reviewer 3 Report

The authors integrated suggested concerns and some comments into the article and provided satisfactory explanations for others not integrated. 

In my opinion, the manuscript is about ready to be published in the Journal.

I still have a little minor "concern" for the authors to address/comment on:

  1. Please, can you support with the numbers of your claims in lines 268 - 274? What is the difference in percentage (%) in power consumption, which generates more dissipation (heat), between your and compared algorithm? 

Author Response

We thank the respected Reviewer for his valuable remarks and comments.

In this regard, we would like to note the following. In our manuscript, we briefly argued the merits of the proposed solutions in terms of their algorithmic structure and provided relevant references. However, the study of implementation features is not the object of our study. We do not set such a goal in this manuscript. As for the power consumption and dissipation, we proceed from the fact that the more the circuit contains logical gates, the greater its power consumption and dissipation. However, the variation of these and other technical and technological parameters is highly dependent on the implementation platform used. Therefore, in this case, there is no sense to talk about any percentages. Moreover, these questions are beyond the scope of our paper. However, we still slightly supplemented the text of the manuscript.

So, we thank the respected Reviewer for his efforts and help in improving the quality of our manuscript.
